# JourneyDB: A Benchmark for Generative Image Understanding

**Keqiang Sun**[1,*]    **Junting Pan**[1,3*,†]    **Yuying Ge**[2],    **Hao Li**[1],    **Haodong Duan**[1],
**Xiaoshi Wu**[1],    **Renrui Zhang**[1],    **Aojun Zhou**[1],    **Zipeng Qin**[1],
**Yi Wang**[3],    **Jifeng Dai**[3],    **Yu Qiao**[3],    **Limin Wang**[3,‡]    **Hongsheng Li**[1,3,4‡]

[1]Multimedia Laboratory, The Chinese University of Hong Kong
[2]The University of Hong Kong    [3]Shanghai Artificial Intelligence Laboratory
[4]Centre for Perceptual and Interactive Intelligence

## Abstract

While recent advancements in vision-language models have had a transformative impact on multi-modal comprehension, the extent to which these models possess the ability to comprehend generated images remains uncertain. Synthetic images, in comparison to real data, encompass a higher level of diversity in terms of both content and style, thereby presenting significant challenges for the models to fully grasp. In light of this challenge, we introduce a comprehensive dataset, referred to as `JourneyDB`, that caters to the domain of generative images within the context of multi-modal visual understanding. Our meticulously curated dataset comprises 4 million distinct and high-quality generated images, each paired with the corresponding text prompts that were employed in their creation. Furthermore, we additionally introduce an external subset with results of another 22 text-to-image generative models, which makes `JourneyDB` a comprehensive benchmark for evaluating the comprehension of generated images. On our dataset, we have devised four benchmarks to assess the performance of generated image comprehension in relation to both content and style interpretation. These benchmarks encompass prompt inversion, style retrieval, image captioning, and visual question answering. Lastly, we evaluate the performance of state-of-the-art multi-modal models when applied to the `JourneyDB` dataset, providing a comprehensive analysis of their strengths and limitations in comprehending generated content. We anticipate that the proposed dataset and benchmarks will facilitate further research in the field of generative content understanding. The dataset is publicly available at https://journeydb.github.io.

## 1   Introduction

In recent times, notable progress has been achieved in the domain of Artificial Intelligence Generative Content (AIGC), particularly in the advancement of diffusion models [1] that have significantly enhanced the quality of generative content. As a consequence, AIGC platforms such as DALLE, Stability AI, Runway, and Midjourney have gained considerable popularity, enabling users to generate exceptionally high-quality images using text prompts composed in natural language. These text prompts encompass both content and style descriptions provided by users, playing a pivotal role in image generation (see Figure 1 for an illustrative prompt). Unlike descriptions acquired from captioning real images, text prompts for image generation tend to exhibit a high level of detail and specificity, surpassing mere portrayal of salient content. The primary objective behind the creation

---

*Equal Contribution
†Project Lead
‡Corresponding Authors

37th Conference on Neural Information Processing Systems (NeurIPS 2023) Track on Datasets and Benchmarks.

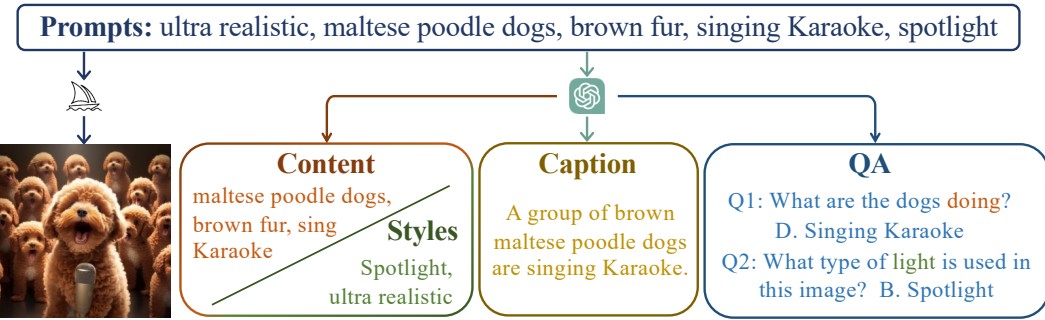

Figure 1: **Data Collection Procedure.** To collect enough generated images, we investigate the Midjourney channel on Discord to collect the available pictures. Then we employ the GPT-3.5 to annotate the downstream tasks, including 1) separating the prompt into "Style" and "Content", 2) generating the caption according to the content words obtained from task 1, 3) generating "Style-relevant questions" and "Content-relevant questions", providing 4 options for each question, together with the answer. Please refer to Section 3 for more details.

of these prompts lies in visual generation, resulting in intricate descriptions that encompass diverse stylistic facets such as lighting, camera angle, artistic style, medium, and more. Moreover, the generated content originates from the users' imagination, often depicting scenes and compositions that are entirely fictional and devoid of real-world existence.

Considering the aforementioned characteristics, we contend that both the elaborate textual prompts and the generated images themselves serve as valuable sources of information that can be incorporated into existing visual understanding benchmarks. On one hand, the detailed text prompts offer a more comprehensive interpretation of the visual scene, enabling us to perceive the scene and comprehend its underlying style. On the other hand, the abundance of novel object compositions in the generated images provides insights into a realm unrestricted by conventional sense biases, facilitating exploration beyond the constraints of traditional visual representations.

Foundation models have achieved unparalleled capabilities across various visual understanding tasks, owing to large-scale pre-training on datasets, such as CLIP [2], Flamingo [3], and BLIP-2 [4]. However, it is essential to acknowledge that current foundation models are primarily pre-trained on real data, giving rise to concerns regarding their generalization ability and effectiveness in handling the distinctive characteristics associated with generative content. These models may not fully capture the nuanced aspects of generative content and might encounter difficulties in comprehending and generating high-quality images based on complex text prompts.

In view of this challenge, our research initiative seeks to address this gap by curating a dataset comprising a substantial number of **4 million** meticulously generated images accompanied by corresponding text prompts. This dataset serves as the fundamental basis for a benchmark consisting of four distinct tasks, which collectively facilitate a comprehensive evaluation of generative content understanding.

The initial task, referred to as **prompt inversion**, involves identifying the text prompts employed by the user to generate the given images. This task serves to decipher the original prompt or description, assessing the model's ability to comprehend both the content and style of the generated images. The second task involves **style retrieval**, wherein the model is tasked with identifying and retrieving similar generative images based on their stylistic attributes. This task evaluates the model's proficiency in discerning subtle stylistic nuances within generative images. The third task centres around **image captioning**, requiring the model to generate descriptive captions that accurately represent the content of the generative image. This task evaluates the model's capability to effectively comprehend and express the visual elements of the generated content using natural language. The fourth and final task is **visual question answering (VQA)**, in which the model is expected to provide accurate answers to questions related to the generative image. This task evaluates the model's ability to comprehend the visual and stylistic content and deliver relevant responses based on the provided questions.

We collected a total of $4,692,751$ pairs of image-text prompts, which were subsequently divided into a training set comprising $4,453,193$ pairs, a validation set comprising $234,156$ pairs, and a test

Table 1: **A comparison between JourneyDB and other commonly-used Text-Image multi-modal datasets.** Among all the commonly-used multi-modal datasets, the proposed dataset is the most versatile, supporting four downstream tasks. H: Human, M: Models

| Dataset | Total Image Num | Label Source | Image Caption | VQA | Prompt Inversion | Style Retrieval |
|---|---|---|---|---|---|---|
| Flickr Caption [9] | 32k | H | ✓ | | | |
| COCO Caption [10] | 164k | H | ✓ | | | |
| VQA v2 [11] | 204k | H | | ✓ | | |
| A-OKVQA [12] | 24k | H | | ✓ | | |
| LAION-COCO [13] | 600M | M | ✓ | | | |
| DiffusionDB [14] | 14M | M | | | ✓ | |
| **Ours** | 4M | H + M | ✓ | ✓ | ✓ | ✓ |

set comprising $5,402$ pairs. We also include $45,803$ images from 22 other text-to-image models provided by HPD v2 [5], including VQ-Diffusion [6], DALL·E 2 [7], StableDiffusion-XL [8], etc., to build the external set for cross dataset evaluation. Given that the generative model is not flawless, some discrepancies in the text prompts may be present. Consequently, for the test set, we carried out human verification, where annotators were tasked with removing word descriptions that do not align with the corresponding images. To create annotations for tasks 2, 3, and 4, we utilized GPT-3.5 to convert text prompts into annotations specific to each task.

To comprehensively evaluate the performance of current state-of-the-art multi-modal models, we conducted extensive assessments using our benchmark dataset. Furthermore, we performed in-depth analyses to gain insights into the strengths and limitations of these models when applied to generative content. Overall, we observed that the state-of-the-art models do not perform as effectively as they do on real datasets, and fine-tuning on the proposed dataset significantly enhances their performance.

In conclusion, our contribution encompasses three key aspects: 1) To the best of our knowledge, we are the first to draw attention to the visual understanding of generated images. 2) We propose `JourneyDB`, a large-scale benchmark that serves as both a training and evaluation resource for this emerging field. 3) We conducted an extensive evaluation of state-of-the-art visual understanding models using the proposed dataset, revealing their relatively limited performance on generative content. We hope that our endeavours will contribute to further advancements in the field of generative content understanding.

## 2 Related Works

### 2.1 Image-Text Datasets

We present a summary of existing image-text datasets in Table 1. The Flickr Caption dataset [9] consists of $32,000$ images obtained from the Flickr [15] platform, accompanied by five reference sentences provided by human annotators. The COCO Caption dataset [10] comprises 164 thousand images, with five independent human-generated captions provided for each image for training and validation, resulting in over $1.5$ million captions. These datasets play a crucial role in fostering the development of the Image-Caption Task. The Visual Question Answering (VQA) v2.0 [11] dataset, which is the second version of the VQA dataset [16], contains open-ended questions about images that require an understanding of vision, language, and commonsense knowledge to answer. A-OKVQA [12], an augmented successor of OK-VQA [17], encompasses a diverse set of 24 thousand questions that demand a broad foundation of common and world knowledge for accurate responses. These datasets involve human employees in the annotation process, ensuring consistently high-quality annotations. However, manual annotation by human annotators is a time-consuming and costly endeavour, thereby limiting the scalability of the datasets. LAION-COCO [13] is another large-scale dataset containing 600 million image-caption pairs, where GPT3.5 is employed to generate more detailed captions. Although these datasets may contain noise due to the cleaning or generation process using pre-trained neural network models, they have demonstrated their utility in training multi-modal models. However, it is important to note that these datasets primarily focus on real images and cater to a specific task. A comparable dataset to the present study is DiffusionDB [18], a large-scale text-to-image prompt dataset comprising 14 million images generated using Stable Diffusion. However, the image quality from Stable Diffusion is plausible, and no further annotations

are available. In this paper, we collect data from Midjourney and provide annotations generated by GPT3.5 to support four downstream tasks.

## 2.2 Text-to-Image Generative Models

Text-to-image generative models [19, 20, 14, 7, 21] aim at generating images according to text conditions, apart from traditional generative models [22, 23, 24, 25], which map random noise to images. Text-to-image generative models have experienced rapid development in recent years, empowering users to create image content through natural language specifications. This field has seen significant progress since Mansimov *et al.*demonstrated that Deep Recurrent Attention Writer (DRAW) can generate images conditioned on text [26, 27]. Since then, several generative architectures and modeling approaches have been applied for text-to-image generation, including autoregressive models [19], GANs [20], and diffusion models [14, 7, 21]. Among these, diffusion models have shown better computational efficiency and the ability to produce higher-quality samples compared to autoregressive models [7]. These diffusion models have reached a level of maturity where they can generate high-quality images suitable for industrial deployment. Notably, Midjourney provides state-of-the-art text-to-image generation service using diffusion models [5]. A vast number of artificial images are generated each day at unprecedented speed. As perception and generation tasks are double sides of the same coin, the achievements in the generative models open new probability for the perception studies. In this context, our dataset aims to organize and consolidate recent progress in text-to-image generative models while laying the foundations for further research in perception studies.

## 2.3 Multi-modal Foundation Models and Datasets

Aided by data from diverse sources, multi-modal foundation models are capable of understanding and connecting data across multiple modalities, such as image, text, audio and so on. As prioneering vision-language models, CLIP [2] and ALIGN [28] adopt contrastive learning paradigms and are pre-trained by millions of web-collected image-text pairs, which showcases promising visual zero-shot capabilities. Flamingo [3] and BLIP-2 [4] further align pre-trained vision backbones with language models with intermediate networks and billions of data pairs, exhibiting superior results on vision-language tasks. OFA [29], Uni-Perceivers [30, 31, 32], and Unified-IO [33] also introduce unified training architectures for different modalities with competitive performance to uni-modal methods. Recently, inspired by the powerful GPT-4 [34], many efforts have been devoted to multi-modal instruction-following models, such as LLaMA-Adapter [35, 36], LLaVA [37] and MiniGPT-4 [38]. Given the textual prompts with image conditions, these models fine-tune a frozen LLaMA [39] to respond to multi-modality instructions, the training data of which is either existing image-caption data [10] or GPT-annotated pairs [37]. Despite the popularity of multi-modal models, it is still rarely explored for their generalization capacity on generated vision-language data, considering the difference between the real-world pre-training data and generative content. In this paper, we propose a large-scale synthetic dataset, `JourneyDB`, along with customized benchmarks to fully validate the extension efficacy current multi-modal models.

## 2.4 Training with Generated Data

It is worth noting that the annotations generated by GPT demonstrate a lower level of noise than expected, validating the effectiveness of these models. Notably, LLaVA [37] introduces a novel instruction-tuning dataset that leverages the capabilities of both GPT3.5 and GPT4. Their experiments reveal a remarkable relative score increase of $295.8\%$, elevating the score from 21.5 to 85.1, thus emphasizing the effectiveness of their generated data. LaCLIP [40] integrates text augmentations by employing text rewriting techniques with GPT3.5 and Bard. By rewriting the textual descriptions within existing image caption datasets, they achieve a notable improvement of $36.08\%$, raising the score from 15.8 to 21.5. StableRep [41] unveils the remarkable potential of using exclusively synthetic data generated from text-to-image models to train highly effective visual representations, surpassing the performance of models trained solely on real image datasets. In a similar vein, VideoChat [42] constructs a dataset by sequentially feeding dense captions to GPT3.5 in temporal order. Despite the inherent challenges in comprehending individual frames with GPT3.5, their successful mastery of understanding the entire video demonstrates the effectiveness of their approach. The generated annotations not only validate the effectiveness of GPT models but also significantly contribute to

Table 2: **Statistics of JourneyDB.** We provide 4 million generated image-prompt pairs, 1 million captions and over 8 million VQA annotations.

| Dataset | Image | Prompt | Labeled Image | Labeled Prompt | Style QA | Content QA |
|---|---|---|---|---|---|---|
| Training Set | 4,453,193 | 1,643,375 | 4,189,737 | 1,385,317 | 7,056,394 | 8,775,971 |
| Validation Set | 234,156 | 82,093 | 234,156 | 82,093 | 311,569 | 374,310 |
| Testing Set | 5,402 | 5,171 | 5,402 | 5,171 | 10,040 | 11,369 |
| External Set | 45,803 | 45,365 | 45,803 | 45,365 | 74,407 | 81,565 |
| Total | 4,738,554 | 1,776,004 | 4,475,098 | 1,517,946 | 7,452,410 | 9,243,215 |

advancing the understanding of images. Therefore, based on our demonstrated results, we firmly believe that our JourneyDB can serve as a valuable tool to enhance numerous image-related tasks.

# 3 Dataset

In this section, we present the methodology employed for dataset collection and annotation, along with relevant statistical insights to gain a deeper understanding of the dataset.

## 3.1 Data Collection

The data collection procedure is presented in Figure 1. In order to obtain a sufficient number of generated images, we investigated the Midjourney channel [43] on the Discord platform [44] to access the available pictures. Within the public Discord channel named "Midjourney," users submit text prompts to the channel, and the Midjourney bot responds with the corresponding generated images. Users then select the preferred image for upscaling, and Midjourney provides the corresponding upscaled images. The chat history contains numerous publicly accessible prompt-image pairs. To collect the data, we utilized DiscordChatExporter [45], a widely used Discord crawler, to download the publicly available images along with their corresponding prompts. In this version of the dataset, we only retained images that were generated solely based on text prompts, filtering out any images conditioned on given images. Additionally, we removed Midjourney-specific arguments, such as "-v 4", to enhance the generalizability of the prompts and ensure their comprehensibility for existing large language models.

Moreover, to improves the diversity of JourneyDB, we additionally introduce another 22 text-to-image generative models into JourneyDB, such as VQ-Diffusion [6], DALL·E 2 [7], StableDiffusion-XL [8], etc., which makes our data a comprehensive benchmark for evaluating the comprehension of generated images. For each generative model, we originally generated $3,200$ images, and a group of $60$ annotators helped clean up the pairs without consistency to obtain the final cross-model test set containing $45,803$ images in total. Please find more details of this part in the appendix D.

## 3.2 Data Annotation

We provide ample annotations for multiple visual understanding tasks. The dataset is compared with existing methods in Table 1, demonstrating its versatility in supporting four downstream tasks.

**Annotation for Visual Understanding.**  In this section, GPT-3.5 is employed to annotate the downstream tasks. Specifically, a set of Midjourney prompts and explicit instructions are provided to GPT-3.5. The objectives are as follows: 1) segmenting the prompt into "Style", "Content", "Atmosphere", and "Others", 2) generating captions based on the content words identified in task 1, 3) generating "Style-relevant questions" and "Content-relevant questions," accompanied by four answer choices for each question. The detailed instructions provided to GPT-3.5 can be found in the Supplementary Materials.

**Clustering of Styles.**  Numerous prompts related to style are highly intricate for style retrieval. Taking inspiration from existing prompt engineering platforms [4], we propose a hierarchical clustering

---

[4]https://www.mbprompt.com/

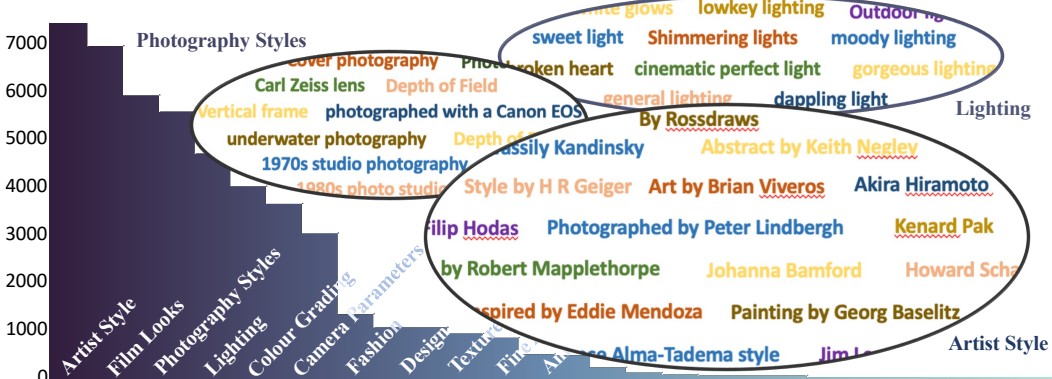

Figure 2: **Distribution and samples of the style prompts.**

approach for organizing styles, which simplifies style retrieval and facilitates user reference. Since traditional word embedding and clustering methods struggle to handle sophisticated style words, we leverage GPT-3.5 for this task. Specifically, we divide the prompts into smaller patches, each comprising 200 prompts, and instruct GPT-3.5 to cluster the words. Subsequently, we manually merge the categories from different patches to obtain the final "style tree". The distribution of the clustered style space is visualized in Figure 2.

**Filtering for Image-Prompt Consistency.** Due to the limitations of Text-to-Image generative models, inconsistencies may arise between the prompt and the generated image. To ensure the quality of the test set, we engaged 40 annotators to identify inconsistent prompt words in the test set. Specifically, given a pair of text prompts and the corresponding generated image, the annotators are instructed to verify if each word is depicted in the image. Words annotated as "Not Appear" are removed to obtain the clean prompts.

### 3.3 Data Statistics

**General Statistics** In this iteration, a total of $4,692,751$ images were collected, all with a resolution exceeding $1024 \times 1024$, accompanied by corresponding text prompts. Among them, $1,730,639$ prompts were found to be independent. Furthermore, $1,472,581$ instances were annotated using the GPT-3.5 model, following the procedure outlined in Figure 1. Additionally, $5,402$ images were filtered out due to inconsistencies between the images and prompts, as determined by the Image-Prompt consistency check. Moreover, $45,803$ images from 22 other text-to-image models provided by HPD v2 [5] are introduced to build the external set for cross dataset evaluation. In addition, a clustering process was conducted to summarize the $70,521$ fine-grained styles into $334$ style categories, displaying a long-tail distribution pattern, as illustrated in Figure 2.

**Dataset Split** Detailed statistics for each subset of the dataset are provided in Table 2. The entire dataset was randomly divided, with approximately a $20 : 1$ ratio, to create the training and validation sets. The training set comprises $4,189,737$ images and $1,385,317$ prompts, while the validation set consists of $234,156$ images and $82,093$ prompts. Additionally, a separate testing set was sampled for manual filtering, consisting of $5,402$ images and $5,171$ prompts.

## 4 Benchmarks

### 4.1 Prompt Inversion

The prompt, which determines both the content and style of a generated image, contains crucial and comprehensive information regarding the image. When presented with an appealing generated image, individuals are eager to discern the prompt employed for its creation. By accurately identifying the prompts, they can further enhance the corresponding image, such as modifying its content or generating images with a similar style.

Table 3: **Evaluation results of Prompt Inversion on JourneyDB.** We list results on the validation set in the upper half, results on the test set in the lower. For all metrics, the higher, the better.

| Models | Validation | | | | | Test | | | | |
|---|---|---|---|---|---|---|---|---|---|---|
| | BLEU-4 | METEOR | ROUGE-L | CIDEr | Similarity | BLEU-4 | METEOR | ROUGE-L | CIDEr | Similarity |
| BLIP-2 OPT [4] | 0.18 | 2.39 | 6.75 | 5.42 | 0.36 | 0.29 | 2.85 | 7.06 | 6.46 | 0.36 |
| BLIP-2 FlanT5 [4] | 0.27 | 2.46 | 7.19 | 6.88 | 0.38 | 0.40 | 2.95 | 7.69 | 8.86 | 0.37 |
| MiniGPT-4 [38] | 1.49 | 5.50 | 12.51 | 10.39 | 0.43 | 1.71 | 6.51 | 13.13 | 11.40 | 0.43 |
| Uni-Perceiver v2 [30] | 0.23 | 2.44 | 9.11 | 12.38 | 0.33 | 0.37 | 2.73 | 9.88 | 15.45 | 0.34 |
| Uni-Perceiver v2$_{FT}$ [30] | **20.6** | **16.9** | **29.1** | **123.2** | **0.59** | **4.68** | **8.56** | **16.98** | **34.01** | **0.51** |

Table 4: **Evaluation results of Image Captioning on JourneyDB.** We list the zero-shot results in the upper half, and the fine-tuned results in the lower. For all metrics, the higher, the better. FT denotes "Fine-Tune".

| Models | Validation | | | | Test | | | | COCO Caption |
|---|---|---|---|---|---|---|---|---|---|
| | BLEU-4 | METEOR | ROUGE-L | CIDEr | BLEU-4 | METEOR | ROUGE-L | CIDEr | CIDEr |
| BLIP-2 OPT [4] | 0.82 | 5.43 | 19.87 | 22.00 | 2.35 | 7.88 | 22.40 | 37.60 | 145.8 (FT) |
| BLIP-2 FlanT5 [4] | 0.54 | 5.02 | 19.94 | 22.18 | 2.07 | 7.62 | **23.12** | **39.62** | 144.5 (FT) |
| Flamingo9B [3] | 0.94 | 6.58 | 14.19 | 10.19 | 1.39 | 6.84 | 17.75 | 19.10 | 79.4 (ZS) |
| MiniGPT-4 [38] | 2.28 | 7.39 | 19.24 | 16.78 | 2.79 | 9.84 | 20.31 | 22.34 | - |
| Uni-Perceiver v2 [30] | 0.41 | 4.50 | 18.72 | 21.88 | 0.94 | 5.21 | 16.71 | 20.13 | 122.5 (FT) |
| Uni-Perceiver v2$_{FT}$ [30] | **8.20** | **12.53** | **27.09** | **50.72** | **3.23** | **10.12** | 22.45 | 31.76 | - |

However, predicting the prompts of an image is a challenging task. Existing visual understanding models, such as image-caption models, often fall short in providing a detailed description of the image's main elements, such as the subject, while neglecting other indispensable details like the viewpoint, illumination, or art style.

Prompt inversion aims to address this gap, involving the process of taking a single image and predicting the corresponding prompts associated with it. We anticipate that the proposed dataset would further facilitate the development of prompt inversion through the in-depth analysis of the prompts.

To evaluate the effectiveness of prompt inversion, we extend the metrics commonly utilized in image captioning, including Bleu, METEOR, ROUGE, and CIDEr. Additionally, we adopt the approach employed in a related Kaggle competition [46] to calculate the Cosine Similarity of the sentence-transformers features [47].

Furthermore, in the supplementary materials, we propose a Question Answering Score (QAS) for evaluating the prompt inversion results. In this paper, we establish a benchmark for the zero-shot prompt inversion task by leveraging state-of-the-art multi-modal models, namely BLIP-2 OPT2.7B [4], BLIP-2 FlanT5XL[4], Flamingo9B[3], MiniGPT-4 [38], and Uni-Perceiver v2 [30]. To ensure optimal performance in this novel task, we customize different prompts for each model.

We evaluate these models on the test set of our dataset, and the results are presented in Table 3. During the experiment, we observed that the existing models struggle to capture the intricate details and style-related information of the input image, leading to lower performance compared to conventional datasets.

To verify the effectiveness of our dataset, we fine-tuned Uni-Perceiver v2 for 20 epochs and noted a significant improvement in the prompt inversion task. It is important to note that we followed the training methodology outlined in [30] without tuning the hyperparameters or utilizing data augmentations. This demonstrates that our `JourneyDB` can complement existing image-text datasets for training prompt inversion models. Nevertheless, it is evident that there is still a substantial way to go in developing a robust and effective prompt inversion model.

## 4.2 Image Caption

Image captioning tasks require multi-modal models to generate textual descriptions for the visual content of an image. In comparison to existing image captioning benchmarks such as COCO Caption [10], `JourneyDB` encompasses both detailed descriptions and high-level summarizations of images, thereby assessing the model's proficiency in fine-grained recognition and holistic understanding.

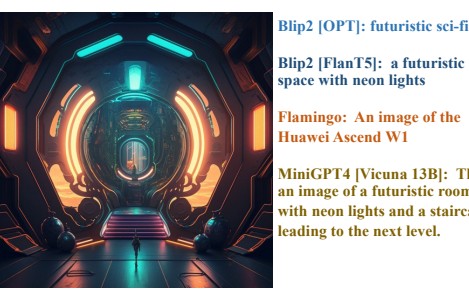

Blip2 [OPT]: futuristic sci-fi art

Blip2 [FlanT5]: a futuristic space with neon lights

Flamingo: An image of the Huawei Ascend W1

MiniGPT4 [Vicuna 13B]: This is an image of a futuristic room with neon lights and a staircase leading to the next level.

(a) GroundTruth: Inside a mother ship from the distant future, a big portal is activated, revealing intricate details and neon lights, all captured in cinematic style with 8k resolution and soft and bold lights.

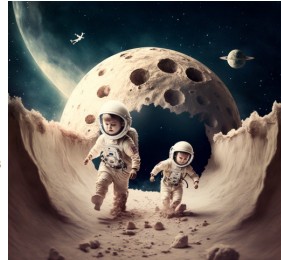

Blip2 [OPT]: two astronauts walking on the moon with a spaceship in the background

Blip2 [FlanT5]: two astronauts walking through a desert with a moon in the background

Flamingo: an image of two astronauts floating in space, with the Earth in the background

MiniGPT4 [Vicuna 13B]: two astronauts walking on the surface of the moon

(b) GroundTruth: A group of kids landing on the moon.

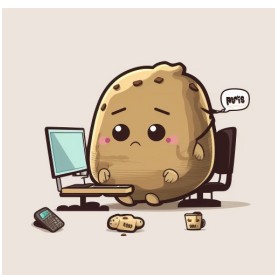

Blip2 [OPT]: a cartoon potato sitting at a computer

Blip2 [FlanT5]: a potato sitting in front of a computer

Flamingo: An image of a cute little robot sitting on a laptop

MiniGPT4 [Vicuna 13B]: The image shows a cartoon potato sitting at a desk with a computer and a calculator in front of it.

(c) GroundTruth: A chibi potato looking sad at work in the office.

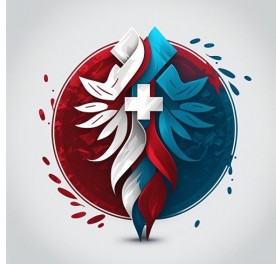

Blip2 [OPT]: the swiss flag with a cross and an eagle on it

Blip2 [FlanT5]: the flag of switzerland with wings

Flamingo: An image of the Swiss flag with the words "Swiss Flag" and "Switzerland". Switzerland is a country in central Europe.

MiniGPT4 [Vicuna 13B]: The flag of Switzerland is a white cross on a red background with a white cross on a red background.

(d) GroundTruth: A nurse logo representing public health in a flat style with silver, blue, and red colors on a pure white background.

Figure 3: **Samples from the validation set of JourneyDB captioning.** The examples show that existing multi-modal models failed to recognize some *key concepts* from the AI-generated images.

Table 5: **Style Retrieval Results**. The metric used there is the Recall.

| Method | Validation | | Test | |
| --- | --- | --- | --- | --- |
| | Over-All | Per-Category | Over-All | Per-Category |
| CLIP-ViT-L/14 [2] | 0.65 | **41.72** | 0.47 | **41.33** |

We evaluate various existing multi-modal models on the image captioning sub-task of JourneyDB. The results are presented in Table 4, indicating the challenges faced by multi-modal models trained on natural images in providing accurate descriptions for AI-generated content. The quantitative performance is notably poorer (significantly worse than COCO Caption results) due to two primary factors: GPT-3.5 tends to generate verbose descriptions for the images in JourneyDB, resulting in lengthy ground-truth captions. This discrepancy between lengthy ground-truth captions and shorter predicted captions undermines the quantitative performance. When describing AI-generated images, the focus may differ in terms of concepts such as emotions, human/object attributes, etc., compared to natural images. However, existing image captioning models have not adequately addressed these concepts.

We provide qualitative examples in Fig 3. Existing multi-modal approaches fail to describe key concepts present in the AI-generated content (e.g., Fig 3(b) depicts *kids* in astronaut suits, Fig 3(d) shows a *sad* potato). Moreover, some models may hallucinate contents that do not exist in the images (e.g., Open-Flamingo hallucinates objects and text in Fig 3(a, c)).

## 4.3 Style Retrieval

We inhabit a captivating world enveloped in a multitude of vibrant colours and ever-shifting illumination. Artists, in turn, develop their distinct styles to convey their unique perspectives of the world. Elements such as weather conditions, moods, and atmospheres all contribute to the style portrayed in an image, resulting in a complex "style system." As detailed in Section 3.2, we have compiled a comprehensive collection of over 150,000 style words to describe the style-related attributes of images.

Table 6: **Evaluation results of the content-relevant and style-relevant zero-shot Multiple-Choice Visual Question Answering on JourneyDB**. The evaluation metric here is accuracy.

| Method | Validation | | Test | |
|---|---|---|---|---|
| | Content | Style | Content | Style |
| Flamingo9B [3] | 32.1% | 31.9% | 35.6% | 41.4% |
| MiniGPT-4 [38] | 28.2% | 26.6% | 31.1% | 29.3% |
| BLIP-2 FlanT5 [4] | 65.8% | 54.9% | 69.7% | 57.4% |

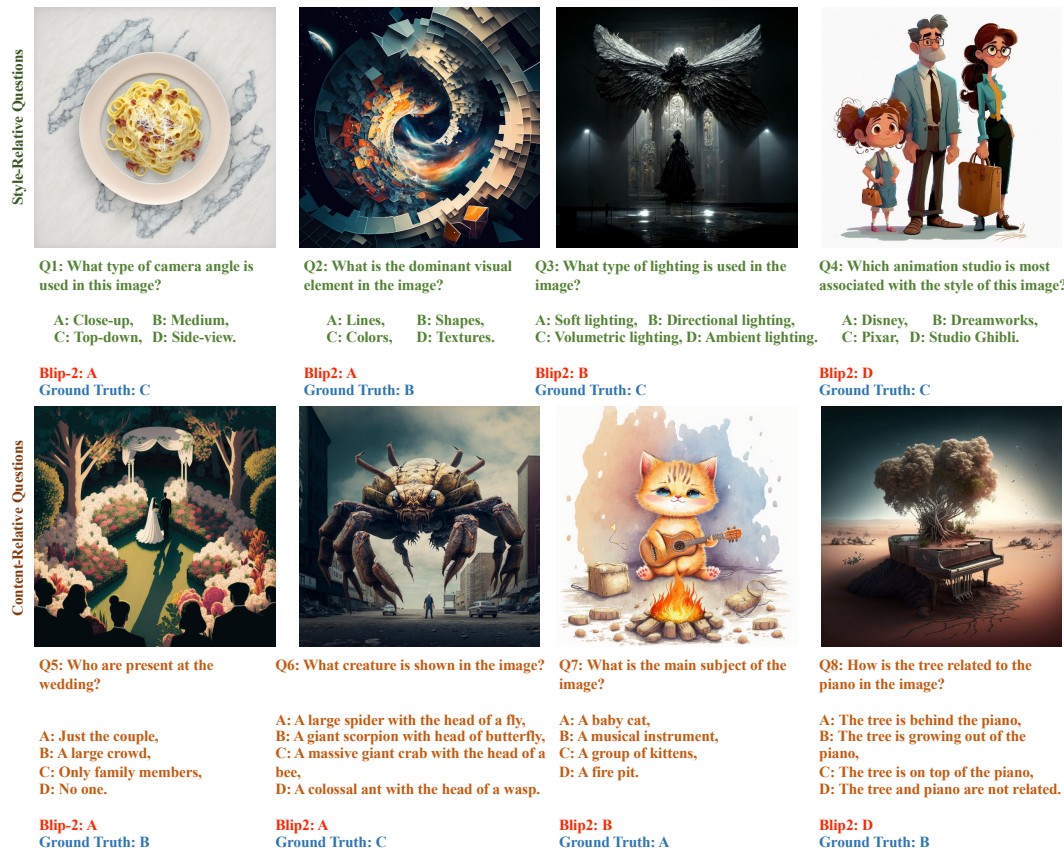

Figure 4: **Failure cases of BLIP-2 [4] for Multiple-Choice Visual Question Answering**.

Given the vast expanse of the style space, identifying the style of a given image poses a significant challenge, even for human observers. Consequently, there is a need for style retrieval techniques to aid in comprehending the style exhibited in an image.

Directly retrieving a style prompt from an extensive pool of candidates is a complex and time-consuming task. Therefore, we employ clustering techniques to group the style prompts into 344 categories, including camera parameters, lighting, artist style, colour schemes, and more, as outlined in Section 3.2. By doing so, we effectively narrow down the search space for style prompt retrieval within each category. To establish a benchmark, we employ CLIP [2] for zero-shot style retrieval evaluation. We extract the features of both the images and the style prompts, subsequently calculating the inner product between the image feature and all candidate style prompts. The results are presented in Table 5. Notably, we observe that conducting retrieval in the overall style prompt space yields significantly low recall. Conversely, the model performs substantially better when performing retrieval in the sub-space of each category.

### 4.4 Visual Question Answering (VQA)

`JourneyDB` comprises a collection of images encompassing abundant and diverse prompts. These prompts not only encompass stylistic attributes but also describe the visual contents of the generated images. To assess the model's proficiency in comprehending both style and content of generative data, we establish two tasks: multiple-choice visual question answering (MC-VQA)[48, 49, 12]. In the MC-VQA tasks, we utilize GPT-3.5 to generate "Style-relevant questions" and "Content-relevant questions" along with three distracting options for each question, in addition to the correct answer. The evaluation metric employed is accuracy, where the model selects its answer from the four options based on the given question and the image. A-OKVQA[12] highlights that MC-VQA overcomes several inherent challenges of direct answer evaluation, such as ambiguity, which is prevalent in open-ended VQA [16, 17]. The versatility of language expression implies that MC-VQA, by directly matching the selected option with the correct answer, provides a lucid and objective evaluation approach. This characteristic proves advantageous, especially considering the extensive spectrum of answers in our benchmark, encompassing a wide range of descriptions for diverse styles and contents.

To assess the performance of current multimodal models in the zero-shot visual question answering task within our benchmark, we adopt a methodology inspired by recent studies [50, 51, 52, 53]. In this approach, we provide the model with a question and its corresponding candidate answers enumerated with symbols ("A", "B", "C", "D"). By assigning the highest probability to a predicted token ("A", "B", etc.), the model selects the associated answer choice as its response.

The evaluation outcomes for the zero-shot multiple-choice visual question answering tasks, specifically the content-relevant and style-relevant tasks, are presented in Table 6. It is evident that the performance of existing multimodal models falls significantly short of satisfactory levels in both the content-relevant and style-relevant MC-VQA tasks. BLIP-2 [4] outperforms Flamingo9B [3] and MiniGPT-4 [38], yet its accuracy remains below $70\%$. These results highlight the substantial challenges that generative data poses to existing models in comprehending the visual contents and stylistic attributes. Generative data often represents scenes and object compositions that are absent in reality, thereby posing difficulties for multimodal models pre-trained on real images to interpret the visual elements of generated images when answering content-relevant questions. For instance, as illustrated in the second row and fourth column of Figure 4, the model fails to recognize the relationship between the piano and the tree in the image and predicts the option "D: the tree and piano are not related." This failure arises due to the rarity of scenes depicting a tree growing out of a piano in the real world. In comparison to answering content-relevant questions, the performance of all models generally deteriorates when addressing style-relevant questions. The generation of multiple-choice questions from text prompts encompassing diverse stylistic aspects, such as camera angle, lighting, and artistic styles, enables a comprehensive evaluation of a model's capacity to identify the stylistic attributes of generative data within `JourneyDB`. However, previous multimodal models are pre-trained using descriptions derived from captioning real images, thereby lacking exposure to the broad range of stylistic variations prevalent in generative data. Consequently, these models encounter difficulties in recognizing and distinguishing the various styles manifested in generative data. As illustrated in Figure 4, BLIP-2 provides incorrect answers to the style-relevant questions in the first row pertaining to camera angle, visual elements, lighting type, and animation style depicted in the images of `JourneyDB`.

## 5 Conclusion

We introduce `JourneyDB`, an extensive benchmark comprising four distinct downstream tasks, aiming to foster advancements in the comprehension of generative content. By providing a platform that facilitates the development of visual understanding in relation to generated images, researchers and practitioners are empowered to drive progress in this field.

## 6 Acknowledgement

Thanks Mengwei R. for her insightful feedback. Thanks Mingjie Z. for his assistance with data curation. This project is funded in part by National Key R&D Program of China Project 2022ZD0161100, by the Centre for Perceptual and Interactive Intelligence (CPII) Ltd under the Innovation and Technology Commission (ITC)'s InnoHK, by General Research Fund of Hong Kong RGC Project 14204021. Hongsheng Li is a PI of CPII under the InnoHK.

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

# Appendices

## A   Data samples

In this section, we randomly sample instances from the dataset to provide a visualization for users. Shown in the Figure 5 are the uncurated samples of JourneyDB. The 5 columns from left to right are images generated by Midjourney, the corresponding prompts, the captions, style-relative and content-relative VQA annotation. We mark the content-relative words with green color, and the style-relative words with orange color.

## B   Details of Data Annotation

In this section, we introduce the details of data annotation, including image-prompt consistency filtering, downstream visual understanding annotations, and style clustering.

### B.1   Image-Prompt consistency filtering

To build a clean test set, we hire a team of 40 professional annotators to mark the prompt word not presented in the corresponding image. Specifically, Given a picture and a corresponding piece of text, the annotators are required to find words or phrases in the text that do not match the picture. Definition of "mismatch": 1) Content that exists in the text but is missing (not reflected) in the picture. For example, if the text is "apples and bananas are placed on a table", and only apples are placed on the table in the picture, then mark "and bananas". 2) It exists in the text, but the content in the picture is inconsistent with the text. For example, if the text is "a red apple is placed on a table", but the picture is a green apple, then mark "red". We show the examples we use in the annotation document in Table 7.

Table 7: Image-Prompt Consistency Filtering Examples.

| Num | Images | Text | Analysis |
| --- | --- | --- | --- |
| 1 | | face portraits of two mechanical cyborg warriors facing off, in the style of a UFC poster, cinematic blue orange lighting, volumetric lighting, smoke, hyper detailed armor, hard surface, neon light tubes, realistic, intricate details, symetrical | All elements are fully represented in the picture. Thus this pair passes review without annotation. |
| 2 | | a cute 1950s style alien with glowing green eyes waving | All elements are well represented in the picture. Thus this pair passes review without annotation. |
| 3 | | Pokémon picnic, detailed, realistic | "Pokémon" are not clearly represented in the picture, so mark it. "picnic", "detailed", and "realistic" are all reflected and not marked. |
| 4 | | sketch, a blue glowing gem with two pieces of rope tied on each side, fantasy, Lord of The Rings, watercolor | "with two pieces of rope tied on each side" it does not conform to the effect shown in the picture, mark it. "Lord of The Rings" does not manifest. "sketch", "a blue glowing gem and fantasy, watercolor" are all reflected and not marked. |

| Images | Prompts | Caption | VQA (Style) | VQA (Content) |
|---|---|---|---|---|
|  | beautiful amanda seyfried as a forest witch with green moss as her dress standing infront of swamp forest facing the camera, potrait view, high detail, water colors, moody atmosphere, style of mignola+diterlizzi+vess | Amanda Seyfried portrays a forest witch in this image, dressed in a gown made of green moss, standing before a swamp forest. She gazes directly at the camera, capturing a portrait view that accentuates her captivating presence. | **Q1: Which artists' style inspired the visual design of this image?**

A: Mignola,
B: Diterlizzi,
C: Vess,
D: None of the above.

**Q2: What type of view does the image provide?**

A: Landscape view,
B: Bird's-eye view,
C: Portrait view,
D: Close-up view. | **Q1: Who is the actress depicted in the image?**

A: Natalie Portman,
B: Emma Watson,
C: Amanda Seyfried,
D: Scarlett Johansson.

**Q2: What is the attire of the forest witch in the image made of?**

A: Green leaves,
B: Flowers,
C: Green moss,
D: Feathers. |
|  | ornate rococo overgrown carousel :: many colorful animals :: wonderland dream nightmare :: photorealistic | An overgrown rococo carousel with many colorful animals in a wonderland dream nightmare, presented in a photorealistic style. | **Q1: What style is the carousel depicted in?**

A: Minimalist,
B: Rococo,
C: Gothic,
D: Baroque. | **Q1: What type of animals are on the carousel?**

A: Birds,
B: Fish,
C: Lions,
D: Horses. |
|  | inside a quantum state | A mesmerizing image capturing the ethereal beauty of quantum particles in a state of superposition, their wave functions delicately intertwined in a mysterious and transcendent dance. | **Q1: What style is depicted in the image?**

A: Impressionism,
B: Realism,
C: Quantum,
D: Surrealism. | **Q1: What is the key concept represented in the image?**

A: Quantum entanglement,
B: Newton's laws of motion,
C: Electromagnetic radiation,
D: Superposition of quantum particles. |
|  | a plate of Chinese and thai food, a bit alien, served on gaiwan porcelain with dragon ornaments, in luxury oriental restaurant | A plate of Chinese and Thai food is elegantly presented on a gaiwan porcelain adorned with dragon ornaments. | **Q1: What type of restaurant is shown in the image?**

A: Italian trattoria,
B: Luxury oriental,
C: Mexican taqueria,
D: French bistro.

**Q2: What is the material of the plate used?**

A: Porcelain,
B: Stoneware,
C: Glass,
D: Melamine. | **Q1: What is placed on the plate?**

A: Chinese and Thai food,
B: Italian pasta,
C: Indian curry,
D: Japanese sushi.

**Q2: What decorative elements can be seen on the porcelain??**

A: Flower patterns,
B: Dragon ornaments,
C: Geometric shapes,
D: Animal prints. |
|  | black pharaoh panther god, crystal purple filigree, insanely detailed and intricate, hypermaximalist, elegant, ornate, hyper realistic, super detailed, 8K | A hypermaximalist, elegant, and ornate 8K image featuring a stunningly detailed depiction of a black pharaoh panther god surrounded by crystal purple filigree. | **Q1: What is the level of detail in the image?**

A: Minimal,
B: Moderate,
C: High,
D: Extreme.

**Q2: How would you describe the style of the image?**

A: Minimalistic,
B: Surreal,
C: Hyperrealistic,
D: Abstract. | **Q1: What color is the filigree in the image?**

A: Gold,
B: Silver,
C: Purple,
D: Black.

**Q2: What animal is depicted as the god in the image?**

A: Lion,
B: Eagle,
C: Panther,
D: Snake. |
|  | 2D retro computer with an astronaut\u2019s inside reaching towards the viewer, hand open, flat illustration, vector art, thick lines, bright colors, simple, black outline --v 4 --q 2 | An astronaut's hand is reaching out from inside a 2D retro computer towards the viewer, hand open, in a flat illustration style with vector art, featuring thick lines, bright colors, and a simple design outlined in black. | **Q1: What type of illustration style is used in the image?**

A: Watercolor,
B: 3D rendering,
C: Vector art,
D: Impressionism. | **Q1: What is the main subject inside the computer?**

A: Robot,
B: Astronaut,
C: Flower,
D: Rocket. |
|  | Film still of rabbit sitting at the counter of an art-deco loungebar, drinking whisky from a tumbler glass, in the style of \"Blade Runner\" (1982), velvety, soft lights, long shot, high quality photo, sharp, look at that detail --v 4 --quality 2 --ar 1:1 | A rabbit sits at the counter of an art-deco loungebar, drinking whisky from a tumbler glass. | **Q1: In which film's style is this image depicted??**

A: Blade Runner (1982),
B: The Great Gatsby (2013),
C: Metropolis (1927),
D: Inception (2010). | **Q1: What is the rabbit doing in the image?**

A: Playing the piano,
B: Drinking whisky,
C: Reading a book,
D: Writing a letter. |

Figure 5: Randomly sampled instances.

## B.2  Visual Understanding Annotation

We employ GPT-3.5 to generate the answer to downstream tasks according to the given prompt. We follow this format to query GPT-3.5:

> *You are a visual art designer. Here is a prompt for a text-to-image generation model: [PROMPT]. You will be required to do the following 3 tasks based on the prompt you receive. Please be faithful to the prompt given to you and do not hallucinate. Directly answer the questions without any redundant sentences in the format of Python dict. The first task is to separate the prompt into 'Style', 'Content', 'Atmosphere', and 'Other' categories. 'Style' words describe the whole image style. 'Content' words describe the image content. 'Atmosphere' words describe the emotional and psychological elements associated with the image, including the mood and feeling conveyed by the scene. If some words are hard to be sorted into 'Style', 'Content', or 'Atmosphere', put them in the 'Other' category. You should try to limit putting words into the 'Other' category as much as possible. The second task is to provide the caption according to the content of the prompt. Only consider the 'Content' words separated in the first task and ignore the 'Style' and 'Atmosphere' words. Be faithful to the 'Content' prompt and do not add any redundant information. The caption should be in a tone that a visual AI assistant is describing the image. The caption should be a single complete sentence. The Third task is to design a set of multiple-choice questions based on the style and content that are separated in the first task. The answers should be in a tone that a visual AI assistant is seeing the image and answering the question. Ask diverse questions and give corresponding answers and also provide wrong options. Only include questions that have definite answers that satisfy the following conditions: 1) one can see the content in the image that the question asks about and can answer confidently, 2) one can determine confidently from the image that wrong options are not in the image. Do not ask any questions that cannot be answered confidently. The answer should not be 'Unknown'. Please include complex questions that are relevant to the content in the image, for example, asking about background knowledge of the objects in the image, asking to discuss events happening in the image, etc. Never ask about uncertain details. Never ask questions you cannot determine from the given prompt. Provide detailed answers when answering complex questions. For example, give detailed examples or reasoning steps to make the content more convincing and well-organized. You can include multiple paragraphs if necessary. Ask at least one question for each category. For each question, there should be 4 options. The options should be relevant but only one is correct. Return a single json file with the following format: [FORMAT]. Strictly follow the provided format please. Directly return the python dict ONLY! Do not say the polite words. Make sure your answer can be parsed by json directly.*

In this way, we encourage the output of the GPT-3.5 can be loaded by the "json" package directly.

## B.3  Style Clustering

Similarly, we ask the GPT-3.5 to cluster the style prompts into several categories. The prompt we use to query GPT-3.5 is:

> *Here are some style-relative prompts: [PROMPTS]. Please help me build a hierarchal tree, to summarise these prompts into several categories like photography styles, camera parameters, colour grading, lighting, film looks, mood, artist style, etc. And each category may have fine-grained sub-categories. Please return a Python dict in the format like: [FORMAT]. You should design the keyword to make sure it summarizes its following list. One prompt can belong to more than one category. Directly return the python dict ONLY! Do not say the polite words. Make sure your answer can be parsed by json directly.*

## C  Additional Experiments

### C.1  Question Answering Score (QAS)

The grammar structure of the prompts is quite different from the caption. The image caption is usually a complete sentence, while some prompts might simply be some broken phases instead. Prior metrics treat the prompts as complete sentences, which do not always hold and bring noise into the evaluation. Therefore, we propose a Question Answering Score (QAS) for the evaluation of the prompt inversion task, which is computed by feeding the predicted prompt to a large language model (LLM) and calculating the accuracy of the LLM answers to the relevant questions provided in the annotation.

Specifically, we make use of Vicuna [54] as the LLM $L$. Given a generated image $I$, a prompt inversion model predicts the prompt $\hat{p}$. And as the annotation, there are $N$ style-relevant questions $q_s$ and answers $a_s$ concerning the style elements in the ground-truth prompts, as well as $M$ content-relevant questions $q_c$ and answers $a_c$ concerning the content. In the following, we do not distinguish between symbols for style and content, but in implementation, we treat them separately to calculate $QAS_s$ and $QAS_c$. We construct a prompt $P$ with $\hat{p}$, $q$, and $a$ in the following format:

> *Here is a prompt: [$\hat{p}$]. Assume you see the image generated from the given prompt. Answer the question: [q]. Choose from: [a]. Directly answer A or B or C or D without any further illustration. Your answer should be one single letter.*

By feeding the constructed prompt $P$ into the LLM $L$ we obtain the predicted result:

$$\hat{a} = L(P). \tag{1}$$

We calculate the average accuracy separately among the $N$ style questions and $M$ content question for this image, and obtain the final QAS by computing the average for all the $K$ images:

$$QAS_s = \frac{1}{K} \sum_K \left( \frac{1}{N} \sum_N \mathbb{I}(\hat{a} = a) \right) \tag{2}$$

and

$$QAS_c = \frac{1}{K} \sum_K \left( \frac{1}{N} \sum_M \mathbb{I}(\hat{a} = a) \right) \tag{3}$$

In this way, we convert the "prompt similarity" problem to compute the accuracy of the question-answering task, which is more interpretable. We show QAS results in the last two columns in Table 8.

Table 8: **Evaluation results of Prompt Inversion on JourneyDB.** We list results on the validation set in the upper half, results on the test set in the lower. For all metrics, the higher, the better.

| Mode | Models | Test | | | | | | |
| --- | --- | --- | --- | --- | --- | --- | --- | --- |
| | | BLEU-4 | METEOR | ROUGE-L | CIDEr | Similarity | $QAS_s$ | $QAS_c$ |
| ZeroShot | BLIP-2 OPT [4] | 0.29 | 2.85 | 7.06 | 6.46 | 0.36 | 12.42% | 18.55% |
| | BLIP-2 FlanT5 [4] | 0.40 | 2.95 | 7.69 | 8.86 | 0.37 | 13.79% | 18.58% |
| | MiniGPT-4 [38] | 1.71 | 6.51 | 13.13 | 11.40 | 0.43 | 17.12% | **26.79%** |
| | Uni-Perceiver v2 [30] | 0.37 | 2.73 | 9.88 | 15.45 | 0.34 | 12.43% | 18.49% |
| Finetune | Uni-Perceiver v2 [30] | **4.68** | **8.56** | **16.98** | **34.01** | **0.51** | **19.71%** | 24.84% |

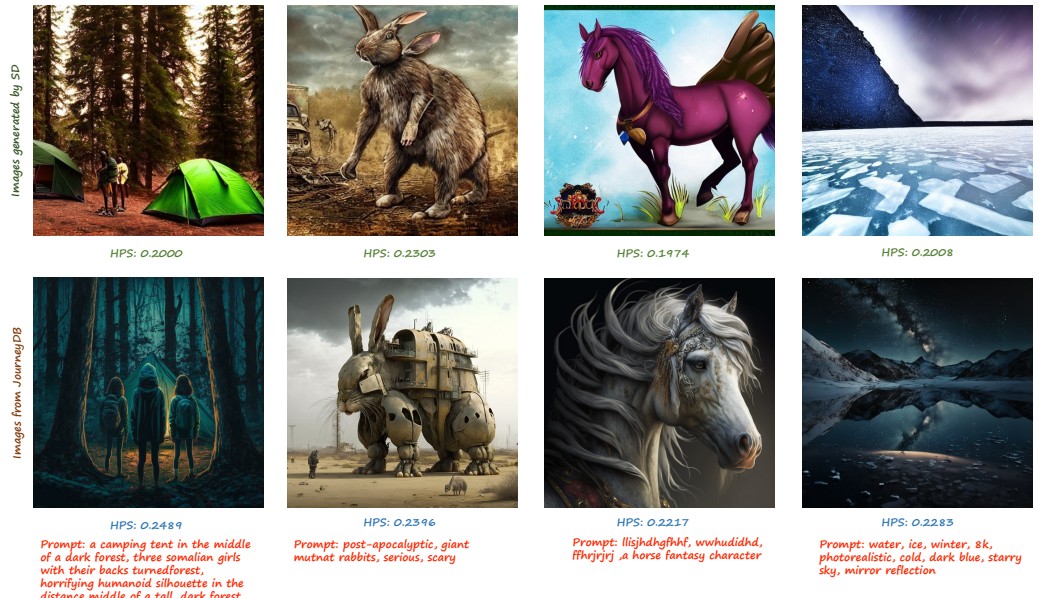

Figure 6: Comparison between images generated by Stable Diffusion v1.4 and images in JourneyDB regarding HPS. Images in the same row are generated with the same prompt.

## C.2 Analysis of Image Quality

As shown in Fig. 6, Images in JourneyDB (credit to Midjourney [55]) exhibit better visual quality than those generated by Stable Diffusion, quantified by Human Preference Score (HPS) [56].

Table 9: **Statistics of JourneyDB.** We provide 4 million generated image-prompt pairs, 1 million captions and over 8 million VQA annotations.

| Dataset | Labeled Image | Labeled Prompt | Style QA | Content QA |
|---|---|---|---|---|
| VQ-Diffusion [6] | 1,888 | 1,869 | 3,067 | 3,346 |
| VQGAN + CLIP [57] | 1,888 | 1,869 | 3,067 | 3,346 |
| GLIDE [58] | 1,898 | 1,878 | 3,101 | 3,349 |
| CogView2 [59] | 1,914 | 1,896 | 3,135 | 3,387 |
| Latent Diffusion [14] | 1,942 | 1,942 | 3,159 | 3,438 |
| Versatile Diffusion [60] | 1,953 | 1,935 | 3,179 | 3,427 |
| Stable Diffusion v1.4 [14] | 2,028 | 2,010 | 3,301 | 3,621 |
| DeepFloyd-XL [61] | 2,052 | 2,028 | 3,334 | 3,655 |
| Epic Diffusion [62] | 2,066 | 2,047 | 3,366 | 3,685 |
| DALL·E mini [63] | 2,097 | 2,075 | 3,415 | 3,739 |
| Dreamlike Photoreal 2.0 [64] | 2,100 | 2,080 | 3,393 | 3,737 |
| Stable Diffusion v2.0 [14] | 2,104 | 2,084 | 3,405 | 3,756 |
| Deliberate [65] | 2,122 | 2,101 | 3,447 | 3,781 |
| LAFITE [66] | 2,124 | 2,105 | 3,439 | 3,804 |
| Realistic Vision [67] | 2,144 | 2,119 | 3,475 | 3,832 |
| FuseDream [68] | 2,176 | 2,157 | 3,536 | 3,893 |
| SDXL Refiner 0.9 [8] | 2,184 | 2,161 | 3,540 | 3,915 |
| MajicMix Realistic [69] | 2,189 | 2,167 | 3,568 | 3,910 |
| ChilloutMix [70] | 2,207 | 2,185 | 3,571 | 3,923 |
| DALL·E 2 [7] | 2,220 | 2,197 | 3,593 | 3,967 |
| Openjourney [71] | 2,237 | 2,214 | 3,630 | 3,992 |
| SDXL Base 0.9 [8] | 2,270 | 2,246 | 3,686 | 4,062 |
| Total | 45,803 | 45,365 | 74,407 | 81,565 |

Table 10: **Evaluation results of Prompt Inversion and Image Captioning on the extension test set of JourneyDB.**

| Models | Prompt Inversion | | | | Image Caption | | | | VQA | |
|---|---|---|---|---|---|---|---|---|---|---|
| | BLEU-4 | METEOR | ROUGE-L | CIDEr | BLEU-4 | METEOR | ROUGE-L | CIDEr | Style | Content |
| BLIP-2 OPT [4] | 3.46 | 8.06 | 24.82 | 51.81 | 3.99 | 9.00 | 26.25 | 55.64 | - | - |
| BLIP-2 FlanT5 [4] | 5.15 | 9.41 | 26.06 | 54.57 | 4.56 | 9.77 | 27.57 | 59.28 | 68.38% | 66.57% |

# D  Cross-model Test Set

As listed in Table 9, we additionally introduce another 22 text-to-image generative models into JourneyDB, such as VQ-Diffusion [6] DALL·E 2 [7], StableDiffusion-XL [8], etc., which significantly improves the diversity of JourneyDB, making it a comprehensive benchmark for evaluating the comprehension of generated images. For each generative model, we originally generated $3,200$ images, and a group of 60 annotators helped clean up the pairs without consistency to obtain the final cross-model test set containing $45,803$ images in total.

We evaluate the BLIP models on the new dataset on the image caption and prompt inversion tasks. Results are shown in Table 10.This additional text-image dataset, with manually cleaned text prompt, image captions, and VQA annotations, serves as a divergent and comprehensive benchmark for the evaluation of the visual understanding model for generated images.

