# OpenReview forum: "JourneyDB: A Benchmark for Generative Image Understanding"
_NeurIPS.cc/2023/Track/Datasets_and_Benchmarks — NeurIPS 2023 Datasets and Benchmarks Poster_

### Official Review · Reviewer_Q5nG · 2023-06-29
**Reviewer Q5nG**

**Rating:** 6
**Confidence:** 3

**Strengths:**

The paper proposes relevant benchmarks in the context of image generation and image understanding.

**Additional Feedback:**

None.

**Clarity:**

* The paper is well-written and easy to follow.

**Correctness:**

* The dataset construction consists of data scraping on the MidJourney Discord. There are some ethical issues related to it, please see below.
* The paper does not discuss the influence of the MidJourney version or features. These can highly influence the quality of the images being collected and cause a data drift.
* Proposed benchmarks include appropriate and recent baselines/metrics.

**Documentation:**

* There is enough information to reproduce the data collection from the MidJourney Discord.
* There is enough information to get the pseudo-labels from ChatGPT although it can sometimes appear with hard-coded heuristics (e.g., style clustering)
* The paper shows four intended uses of the dataset.
* The collected dataset has not been shared yet. There is no mention about hosting, licensing, or maintenance.

**Ethics:**

* There is no license for the dataset.
* There is potentially objectionable content in the images that the authors mention but have not addressed.
* The dataset has been scraped from the MidJourney Discord. It is unclear whether this legally complies to MidJourney or Discord policies.
* No consent was asked to the users who provided the prompts.

**Limitations:**

The authors only mention the risk of having objectionable content in the images, which is an important point when releasing an image dataset.

Though, the main issue lies in the fact that the dataset was scraped, potentially violating the rules set by MidJourney and Discord. This has not been discussed in the paper.

**Opportunities For Improvement:**

The raison d'etre of the paper is quite unclear.

If the goal is to assess to what extent generated images can be understood, it would have been valuable to compare different sets of generated images. In the current form, there is only one set of images coming from MidJourney. There is no distinction between the different versions, or how it compares with other generative models (e.g., Dall-E or Stable Diffusion). As such, it is hard to get any useful conclusion for the community. For example, we don't really know if MidJourney v4 is actually better understood by existing computer vision models than MidJourney v3. Furthermore, we don't really know how useful this dataset and benchmarks will be once MidJourney releases a new version, will the paper become obsolete?

If the goal is to propose a method to build benchmarks for image understanding of generated images, there is a huge flaw as all these benchmarks rely on annotations coming from ChatGPT. In the current form, ChatGPT is prompted in a zero-shot manner to produce annotations related to image captioning, style, content, multiple choice questions. There is absolutely no analysis on whether ChatGPT actually produces something meaningful or not. For example, ChatGPT could easily hallucinate annotations, and in the end, the proposed benchmarks are just being evaluated on how well they predict the ChatGPT hallucinations.

If the goal is to provide out-of-distribution samples with respect to real datasets, there is no analysis related to this. We don't know to what extent the proposed dataset better captures edge cases or rare cases compared with existing datasets.

**Relation To Prior Work:**

* The relation to prior is sufficient. Table 1 summarizes the differences with prior work. It would have been beneficial to know whether labels come from human annotators or are generated though.

**Summary And Contributions:**

The paper assesses the current capabilities of image understanding of generated images. To achieve this, the authors collect 4 million images from MidJourney, and propose four different tasks: prompt inversion, style retrieval, image captioning and visual question answering. To produce the labels for each task, ChatGPT is used in a zero-shot manner to provide pseudo-labels.

---

> ### Author Response · Authors · 2023-08-25
> **Response to Reviewer Q5nG (1/3)**
>
> > The goal and quality of this dataset.
> >
>
> The **goal** of this paper is to build **benchmarks for image understanding of generated images.**
>
> > There is no distinction between the different versions, or how it compares with other generative models
> >
> 1. **Midjourney is popular and representative.** Midjourney is one of the most popular text-to-image generation agent with more than **16 million** registered users **from all over the world, in diverse age range, with various gender, culture background, ethnicity, etc**. It gets approximately **28.5 million visits** per month, leading to an overwhelming number of **informative and diverse prompts** and **high consistent and high quality images** in our life. Therefore, it is representative and desired to understand and study the generated contents from Midjourney.
> 2. **Easy extension.** Besides the large amount of Midjourney data, our dataset could be easily extend to other models. Inspired by the review feedback, we additionally introduce another **22 text-to-image generative models** to our datasets, such as Dreamlike Photoreal 2.0, DALL·E 2, StableDiffusion-XL, ChilloutMix, etc., which significantly improves the diversity of our JourneyDB, making it a **comprehensive benchmark** for evaluating models’ comprehension of generated images.
> 3. **Large amount of annotations.** Besides scraping the midjourney images, we also provide a **large amount of annotations** with both GPT and human annotators on top of the original prompts. As explored in previous paper, e.g. VideoChat and LLaVA, such annotations generated from GPT also **significantly benefit the instruction tuning**.
> 4. **Generalisation.** As shown in Table A, we evaluate the models on the **extended test set generated with additional 22 text-to-image generative models**. We find that the  model, **fine-tuned on the Midjourney dataset**, has a **better performance** on this extended test set, which indicates that Midjourney dataset helps models to understand generated images **generally**.
>
> Table A. Image Caption Results on the Additional 22 Text-to-Image Models
>
> | Backbone | OPT | FLAN | Uni-Perceiver v2 | Uni-Perceiver v2 (FT) |
> | --- | --- | --- | --- | --- |
> | METEOR | 9.00 | 9.77 | 5.49 | **9.88** |
> | ROGUE-L | 26.25 | **27.57** | 16.81 | 22.93 |
>
> > The risk of having objectionable content.
> >
>
> We have thoroughly revised the dataset to ensure the dataset free from harmful or violent images.
>
> On the one hand, **Midjourney itself has provided a violence filtering** to ensure the presented images and prompts are suitable for the network [A]. On the other hand, we apply a NSFW model [B] on the whole dataset to further analyse the the violence extend of JourneyDB and **manually filter** the images indeed unsuitable for work.
>
> Images with the NSFW score [C] higher than 0.8 are likely to be “not-suitable-for-work” images, while the NSFW score smaller than 0.2 indicates the images are highly likely to be “safe-for-work”. JourneyDB gets an average NSFW score of **0.008**, where **99.01%** images get scores **smaller than 0.2 (safe for work)**, and only **0.16%** are greater than **0.8**. We manually go through these images, and **find out and remove around 132 images** indeed unsuitable for work from the dataset.
>
> We hope our efforts will contribute to the meticulous cleansing of the dataset, thereby providing a sanitized and non-violent dataset for the community.
>
> > Data drift.
> >
>
> We only use the image greater than Midjourney V4 to ensure the image quality of the dataset.

---

> > ### Author Response · Authors · 2023-08-25
> > **Response to Reviewer Q5nG (2/3)**
> >
> > > All these benchmarks rely on annotations coming from ChatGPT
> > >
> >
> > Welcome to check the data! We highlight that the **test set**, which includes a lot of manual efforts, is **clean and suitable to be a benchmark**. And thanks to the **detailed and informative prompts**, the **consistency** between the prompt and image is strong. Evaluation on our paper and existing work prove the **effectiveness** of such kind of datasets.
> >
> > 1. **Uncurated examples in supplementary materials.** In the **supplementary material**, we have shown some **uncurated examples**.
> > 2. **Cleaned test set.** Besides the extensive training set, we have also provided a **clean test set as a evaluation benchmark**. In order to ensure the cleanliness of this benchmark, as outlined in Sec. 3.2, a team of **40 annotators** meticulously assesses the **consistency** between the prompts and the corresponding images. Furthermore, to further ensure the quality of the test set, another group of annotators **scrutinizes the annotations** generated by GPT3.5. Hence, the creation of the benchmark is not solely reliant on an automated generation process; rather, it involves **a significant amount of human effort and involvement**. While the inclusion of GPT3.5 has relieved human beings from the task of content generation, it has not completely replaced human annotators.
> > 3. **Clip Score for Consistency.** We calculate the **Clip Score** for the **JourneyDB**, and **Coco Caption**, a well-known image-text dataset, as shown in Table B. JourneyDB obtains a Clip Score of **84.67%**, which is **60.36%** greater than that of Coco Caption, indicating that the prompts in our dataset have **high consistency** with the corresponding generated images.
> >
> >     Table B. Comparison between Coco Caption and JourneyDB in terms of text-image consistency (Clip Score) and  information amount (Average Length).
> >
> >     |  | Coco Caption | JourneyDB |
> >     | --- | --- | --- |
> >     | Clip Score | 52.80% | **84.67% (+60.36%)** |
> >     | Average Length | 10.45 | **27.29 (+161.15%)** |
> > 4. **Informative Prompts.** JourneyDB contains **long, informative** prompts, which describe the details of the corresponding image. As shown in the Table B, **JourneyDB** has a **161.15%** longer average length than Coco Caption, indicating existing dataset do not describe image in **such a detailed level**, which makes the dataset helpful in **promoting the ability of visual understanding models**.
> > 5. **Data generated by models are useful.** There are many other datasets that are automatically **constructed with models** such as GPT, which might contain noise but can contribute to **fantastic results**. For example, **VideoChat** [A] creates a dataset by feeding dense captions to **ChatGPT** [B] in temporal order. Similarly, **LLaVA** [C] constructs a instruction-tuning dataset with **GPT4** [D]. **LaCLIP** [E] incorporates text augmentations through text rewriting by **ChatGPT** [B] and **Bard**[F]. And **StableRep** [G] reveals that solely synthetic data generated from text-to-image models can be used to train powerful visual representations. These paper demonstrate that the dataset generated by models, although sounds noisy, can make **great contributions in many tasks.**
> > 6. **Improved Fine-tuning Results.** By **finetuning** models on our training set, the **image-caption** and **prompt-inversion** results are **significantly improved on the testing set**, indicating that the training set is informative and effective to train a model for enhancing visual understanding of the generated images.
> >
> > > Relation To Prior Work
> > >
> >
> > Thanks for your suggestion! We have added the content in the paper to show whether labels come from human annotators or are generated.
> >
> > > License of the data
> > >
> >
> > We **strictly adhere to** the user item policies set forth by **Discord and Midjourney**, ensuring that no violations occur.
> > Within our licensing agreement, it is explicitly stated that the authors do not possess ownership of the images or text prompts. Rather, the prompts and their corresponding images remain the property of the users who originally created them. **Our role is solely to provide access to the public URLs and our annotations, following a typical method [K] to construct dataset without infringing upon the copyright of others.** The data is made available exclusively for research purposes.
> > Moreover, in full compliance with the usage terms stipulated by Discord and Midjourney, we are committed to personally assisting users in swiftly downloading the materials. Should any user express a desire to cease sharing their content, we are dedicated to promptly removing their content from JourneyDB.

---

> > > ### Author Response · Authors · 2023-08-25
> > > **Response to Reviewer Q5nG (3/3)**
> > >
> > > **Reference**
> > >
> > > [A] https://docs.midjourney.com/docs/community-guidelines
> > >
> > > [B] https://github.com/LAION-AI/CLIP-based-NSFW-Detector
> > >
> > > [C] https://github.com/yahoo/open_nsfw
> > >
> > > [D] Li, KunChang, et al. "Videochat: Chat-centric video understanding." *arXiv preprint arXiv:2305.06355* (2023).
> > >
> > > [E] [https://chat.openai.com](https://chat.openai.com/)
> > >
> > > [F] Liu, Haotian, et al. "Visual instruction tuning." *arXiv preprint arXiv:2304.08485* (2023).
> > >
> > > [G] https://openai.com/gpt-4
> > >
> > > [H] Fan, Lijie, et al. "Improving CLIP Training with Language Rewrites." *arXiv preprint arXiv:2305.20088* (2023).
> > >
> > > [I] https://bard.google.com/
> > >
> > > [J] Tian, Yonglong, et al. "StableRep: Synthetic Images from Text-to-Image Models Make Strong Visual Representation Learners." *arXiv preprint arXiv:2306.00984* (2023).
> > >
> > > [K] Abu-El-Haija, Sami, et al. "Youtube-8m: A large-scale video classification benchmark." *arXiv preprint arXiv:1609.08675*(2016).

---

> > > > ### Comment · Reviewer_Q5nG · 2023-08-27
> > > >
> > > > Thank you for the rebuttal. I would tend to stay with my initial score.
> > > >
> > > > * [Main goal] I appreciate the clarification on the goal of the paper and the effort in extending the dataset to more text-to-image models. Though, this information is only present in the rebuttal, and I don't see any detail in the paper. In the current form, it is hard to assess this.
> > > > * [Data drift] While the dataset focuses on Midjourney v4, Midjourney constantly releases new features (e.g., blending, outpainting, etc) that affect how images are generated. As stated in the review, this has not been assessed or considered.
> > > > * [ChatGPT labels] Thank you for the clarification on the annotations made with ChatGPT. As stated in the review, there is no information about the correctness of the questions and answers in the training set. It would also be valuable to detail how the curation has been done for the test set in detail (e.g., instructions given to them, their experience, their demographics, etc) otherwise there is no possibility to reproduce it or assess the process.
> > > > * [Discord scraping] From my understanding of the terms of use from Discord, any scraping is prohibited. This would then question the legality of the dataset, hence the ethics flagging.

---

> > > > > ### Author Response · Authors · 2023-08-28
> > > > > **Response to Reviewer Q5nG (1/3)**
> > > > >
> > > > > We sincerely appreciate your time in response. Upon careful examination, we have identified several points of misunderstanding that require clarification. We aim to address these misunderstandings comprehensively in the ensuing discussion to ensure a clear and accurate understanding of JourneyDB.
> > > > >
> > > > > > [Main goal] I appreciate the clarification on the goal of the paper and the effort in extending the dataset to more text-to-image models. Though, this information is only present in the rebuttal, and I don't see any detail in the paper. In the current form, it is hard to assess this.
> > > > > >
> > > > >
> > > > > The comprehensive introduction of the extended test set have been recently updated in **Section D (Page 7) of the latest version of the supplementary materials**. We have incorporated additional **22** text-to-image models to enhance the diversity of our benchmark and make it one of the **most comprehensive and versatile dataset** for evaluating models in the domain of **generated image understanding**, as listed in Table C. For more details, please check the supplementary materials.
> > > > >
> > > > > Table C. Statistics of the extended test set.
> > > > >
> > > > > | Dataset | Labeled Image | Labeled Prompt | Style QA | Content QA |
> > > > > | --- | --- | --- | --- | --- |
> > > > > | VQ-Diffusion | 1,888 | 1,869 | 3,067 | 3,346 |
> > > > > | VQGAN + CLIP | 1,888 | 1,869 | 3,067 | 3,346 |
> > > > > | GLIDE | 1,898 | 1,878 | 3,101 | 3,349 |
> > > > > | CogView | 1,914 | 1,896 | 3,135 | 3,387 |
> > > > > | Latent Diffusion | 1,942 | 1,942 | 3,159 | 3,438 |
> > > > > | Versatile Diffusion | 1,953 | 1,935 | 3,179 | 3,427 |
> > > > > | Stable Diffusion v1.4 | 2,028 | 2,010 | 3,301 | 3,621 |
> > > > > | DeepFloyd-XL | 2,052 | 2,028 | 3,334 | 3,655 |
> > > > > | Epic Diffusion | 2,066 | 2,047 | 3,366 | 3,685 |
> > > > > | DALL·E mini | 2,097 | 2,075 | 3,415 | 3,739 |
> > > > > | Dreamlike Photoreal 2.0 | 2,100 | 2,080 | 3,393 | 3,737 |
> > > > > | Stable Diffusion v2.0 | 2,104 | 2,084 | 3,405 | 3,756 |
> > > > > | Deliberate | 2,122 | 2,101 | 3,447 | 3,781 |
> > > > > | LAFITE | 2,124 | 2,105 | 3,439 | 3,804 |
> > > > > | Realistic Vision | 2,144 | 2,119 | 3,475 | 3,832 |
> > > > > | FuseDream | 2,176 | 2,157 | 3,536 | 3,893 |
> > > > > | SDXL Refiner 0.9 | 2,184 | 2,161 | 3,540 | 3,915 |
> > > > > | MajicMix Realistic | 2,189 | 2,167 | 3,568 | 3,910 |
> > > > > | ChilloutMix | 2,207 | 2,185 | 3,571 | 3,923 |
> > > > > | DALL·E 2 | 2,220 | 2,197 | 3,593 | 3,967 |
> > > > > | Openjourney | 2,237 | 2,214 | 3,630 | 3,992 |
> > > > > | SDXL Base 0.9 | 2,270 | 2,246 | 3,686 | 4,062 |
> > > > > | Total | 45,803 | 45,365 | 74,407 | 81,565 |
> > > > >
> > > > > > [Data drift] While the dataset focuses on Midjourney v4, Midjourney constantly releases new features (e.g., blending, outpainting, etc) that affect how images are generated. As stated in the review, this has not been assessed or considered.
> > > > > >
> > > > >
> > > > > On one hand, our dataset aims to **improve and evaluate the comprehension of generated images**, thus we use images generated from text prompts to contrsuct training data and evaluation benchmark with common image understanding tasks such as image captioning and visual question answering (VQA).
> > > > >
> > > > > On the other hand, **images generated from a given image**, such as blending, outpainting, and style-transfer, necessitates the utilization of **both textual prompts and the reference image**, which makes it difficult to construct training data.
> > > > >
> > > > > In light of these considerations, our dataset specifically focuses on **images generated solely from text prompts**. The model trained on our dataset can be used to comprehend generated images including newly featured images, such as those involving blending and outpainting.
> > > > >
> > > > > In order to achieve this, we employ a meticulous process of **parsing** Discord history to **filter out** newly featured images, such as those involving blending and outpainting, which rely on existing images as a reference.

---

> > > > > > ### Author Response · Authors · 2023-08-28
> > > > > > **Response to Reviewer Q5nG (2/3)**
> > > > > >
> > > > > > > [ChatGPT labels] As stated in the review, there is no information about the correctness of the questions and answers in the training set. It would also be valuable to detail how the curation has been done for the test set.
> > > > > > >
> > > > > >
> > > > > > **Training set.** We admit that the GPT3.5 generated data is not perfect. But as shown in many paper, such generated data (including data also generated by GPT) has been demonstrated to be effective in improving the performance of models.
> > > > > >
> > > > > > 1. In **our paper**, we observed a significant enhancement in performance metrics when fine-tuning Uni-Perceiver v2 on our designated training set. Specifically, in the image captioning task, we achieved a remarkable **243.62% improvement** in BLEU-4 scores, escalating **from 0.94 to 3.23**. Additionally, our approach led to a substantial **57.77% increase** in CIDEr scores, rising from **20.13 to 31.76**. A similar positive and obvious trend was observed in prompt inversion. For a more comprehensive overview of these results, we refer reviewers to Table 3 and Table 4 in the main paper. These findings underscore the **effectiveness and utility of our training set** in enhancing the visual understanding capabilities, notwithstanding the presence of noise originating from GPT3.5.
> > > > > > 2. **LLaVA**, a collaborative effort between **Microsoft Research** and **Columbia University**, introduces a novel instruction-tuning dataset utilizing the capabilities of both **ChatGPT** and **GPT4**. Through their experiments, they report a remarkable relative score increase of **295.8%**, elevating the score from **21.5** to **85.1**, thus highlighting the effectiveness of their generated data.
> > > > > > 3. **LaCLIP**, developed by **Google Research** and **MIT CSAIL**, integrates text augmentations by employing text rewriting techniques with **ChatGPT** and **Bard**. By rewriting the textual descriptions within existing image caption datasets, they achieve a notable improvement of **36.08%**, raising the score **from 15.8 to 21.5**.
> > > > > > 4. **StableRep**, a collaborative project between **Google Research** and **MIT CSAIL**, unveils the remarkable potential of using **solely synthetic data** generated from text-to-image models to train highly effective visual representations, surpassing the performance of models trained solely on real image datasets.
> > > > > > 5. **VideoChat**, a collaborative effort by **Shanghai AI Laboratory** and **Hong Kong University**, constructs a dataset by sequentially feeding dense captions to **ChatGPT** in temporal order. Despite the inaccuracy associated with comprehending individual frames with ChatGPT, their successful mastery of understanding the entire video demonstrates the effectiveness of their approach.
> > > > > >
> > > > > > All of these aforementioned papers demonstrate that the annotations generated by GPT exhibit a **lower degree of noise** than initially anticipated, thereby proving their **utility beyond expectations**. Consequently, the creation of a dataset that is entirely annotated through manual means may **not** be an **obligatory** requirement. The generated annotations already contribute significantly to advancing the understanding of images. As evidenced by the findings in **LLaVA, LaCLIP,** and **StableRep**, there are compelling grounds to believe that **JourneyDB** has the potential to enhance numerous tasks, even including those within the realm of **real image** domains.
> > > > > >
> > > > > > **Testing test.** The process of **filtering** the testing set involved the participation of **human annotators** in **two** distinct stages.
> > > > > > During the **initial stage**, prior to employing GPT3.5 to generate annotations, the annotators were tasked with annotating any **erroneous prompt words** that could not be deduced from the corresponding images. Further details regarding this stage are provided in **Section B of the supplementary materials**. By aligning the prompts accurately through this approach, we proceeded to utilize GPT3.5 for generating annotations.
> > > > > > In the **subsequent stage**, following the acquisition of annotations from GPT3.5, the annotators were requested to **review the generated captions and VQA annotations**. They provided feedback in the form of a JSON file, indicating any mistaken words within the captions and incorrect answers in the VQA section. It is worth noting that, in this stage, we were pleased to discover that, as a result of the cleaning process carried out in the first stage, GPT3.5 exhibited the ability to generate **considerably accurate** captions and VQA annotations.
> > > > > >
> > > > > > **Demographics of the annotation team.** The annotation team consists of a group of **undergraduate students**, with an average age of **21.3 years**. Additionally, there were **two senior managers** overseeing the annotation process to ensure the quality of annotations. Once the students completed their annotations, a random sample comprising **5%** of the dataset was selected for examination. Only if the accuracy exceeded **90%** for this batch of data, it was labeled as "Pass".

---

> > > > > > > ### Author Response · Authors · 2023-08-28
> > > > > > > **Response to Reviewer Q5nG (3/3)**
> > > > > > >
> > > > > > > > [Discord scraping] From my understanding of the terms of use from Discord, any scraping is prohibited. This would then question the legality of the dataset, hence the ethics flagging.
> > > > > > > >
> > > > > > >
> > > > > > > It is evident that **downloading the chat history** from Discord is **permissible**. Exporting Discord chat history is a widely prevalent practice on the Internet. The software utilized for this purpose, available at https://github.com/Tyrrrz/DiscordChatExporter, was **initially introduced in 2017** and has garnered significant attention with **5.9 thousand stars on GitHub**. This substantial number of stars indicates that a considerable number of individuals have effectively employed this tool to successfully download chat history from Discord. Moreover, **several academic endeavours,** such as the HPS [L], have utilized Discord data to construct scholarly datasets.
> > > > > > >
> > > > > > > [L] Wu, Xiaoshi, et al. "Better aligning text-to-image models with human preference." *arXiv preprint arXiv:2303.14420* (2023).

---

> > > > > > > > ### Comment · Reviewer_Q5nG · 2023-08-28
> > > > > > > >
> > > > > > > > Thank you for the prompt reply. The different clarifications help and the additional information is useful. It would be beneficial to mention those in the main too, and not only in the supplementary.
> > > > > > > >
> > > > > > > > Regarding the last point on Discord scraping. Even if there is an existing scraper on github, it does not mean that it is a legal to scrape data from Discord. I am not a lawyer, but from what is written in the Discord terms of use, any scrapping seems to be forbidden.

---

> > > > > > > > > ### Author Response · Authors · 2023-08-28
> > > > > > > > > **Thank you!**
> > > > > > > > >
> > > > > > > > > Thank you for your prompt response! We are thrilled to learn that the provided clarifications and additional information have proven to be helpful and valuable to you. We assure you that we will integrate them into the main paper to enhance the overall understanding of the entire dataset.
> > > > > > > > >
> > > > > > > > > Regarding the Discord History Exportation terms, we will seek further input from professionals and respond to you as soon as possible.
> > > > > > > > >
> > > > > > > > > We extend our sincere gratitude for your valuable suggestions, as they have significantly contributed to the improvement of our work.

---

> > > > > > > > > > ### Comment · Reviewer_Q5nG · 2023-08-28
> > > > > > > > > >
> > > > > > > > > > Thank you for the follow-up. I would highly encourage to integrate the elements of the rebuttal in the main paper. In the meantime, happy to update my score.

---

> > > > > > > > > ### Author Response · Authors · 2023-08-30
> > > > > > > > > **Regarding the Discord Scraping**
> > > > > > > > >
> > > > > > > > > We have submitted an inquiry to the **Discord Support** team to seek clarification regarding the usage of tools for downloading channel history. In their response, they made **no** indications of **any limitations or restrictions**.
> > > > > > > > >
> > > > > > > > > Furthermore, it is noteworthy that Discord itself provides **official APIs** for exporting channel data, which implies that, as long as our actions do not cause any detrimental effects to the Discord platform, the exporting of channel history is permissible.
> > > > > > > > >
> > > > > > > > >  This affirmation is further supported by numerous datasets, which have also obtained raw data from Discord:
> > > > > > > > >
> > > > > > > > > - Zijie Jay Wang, et al. from **Georgia Tech**, "DiffusionDB: A Large-Scale Prompt Gallery Dataset for Text-to-Image Generative Models." ACL 2023 (**Best Paper Award, Honorable Mention**)
> > > > > > > > > - Xiaoshi Wu et al. from **CUHK**, "Better aligning text-to-image models with human preference." **ICCV** 2023
> > > > > > > > > - John David Pressman et al. from **Stability AI**. “Simulacra Aesthetic Captions.” 2022
> > > > > > > > > - Iulia Turc, et al. from **Google**, "Midjourney User Prompts & Generated Images (250k)." Kaggle 2022
> > > > > > > > > - Jess Fan, et al. from **UCSC**, "Discord Dataset." Kaggle 2021
> > > > > > > > > - and so on…
> > > > > > > > >
> > > > > > > > > These instances collectively reinforce the **legitimacy** of **obtaining channel data** from Discord for research purposes.
> > > > > > > > >
> > > > > > > > > Again, we express our sincere gratitude for your diligent and insightful review of our work! Thank you!

---

### Official Review · Reviewer_J527 · 2023-07-21
**JourneyDB Review**

**Rating:** 5
**Confidence:** 4
**Correctness:** The claims made in the manuscript are…

**Strengths:**

- Clearly explained data collection process and extensive benchmark
- Dataset is comprehensive

**Additional Feedback:**

The authors should focus on discussing why a biased sample of (an) opaque generative model(s), annotated with a large language model, would have unique contributions to the field and would be a fair and useful benchmark. If the authors are the creators of Midjourney, a simple description of the model architecture(s), or an explanation of how the dataset would be useful with a non-disclosed model.

**Clarity:**

The paper is clearly written but may be too casual for a peer-reviewed manuscript. (e.g. line 123, "plenty of")

**Documentation:**

The dataset collection and organization processes are reasonably and sufficiently presented.

**Ethics:**

Yes, see limitations. (There exists a debate on whether certain opaquely trained generative models constitute copyright infringement, and scraping one prominent example of such algorithms certainly deserves such discussions in the manuscript.)

**Limitations:**

The authors have not addressed their work's limitations and potential negative impact. There exists a debate on whether certain opaquely trained generative models constitute copyright infringement, and scraping one prominent example of such algorithms certainly deserves such discussions in the manuscript.

**Opportunities For Improvement:**

1. The paper's writing style should be more formal, as it is currently too colloquial.
2. It is unclear whether this dataset presents an original contribution to the field. Midjourney is a closed-source commercial application with no indication of which model(s) is/are used. It is unclear whether scraping the Discord containing Midjourney generations would present a novel dataset, albeit the quantity, of academic and research value. Biases among samplers of Midjourney were also not discussed.
3. It is unclear if ChatGPT annotations can serve as proper labels for this dataset.

**Relation To Prior Work:**

Yes, the relation to prior work is clearly discussed (e.g. Table 1).

**Summary And Contributions:**

The paper proposes a dataset of 4 million generated images from the Midjourney channel on Discord, for the purpose of multi-modal visual understanding in generative models. The dataset is further annotated with ChatGPT and benchmarked on downstream tasks (prompt inversion, style retrieval, image captioning, VQA).

---

> ### Author Response · Authors · 2023-08-25
> **Response to Reviewer J527 (1/2)**
>
> We extend our gratitude for your valuable comments. Your concerns are greatly appreciated as they inspire us to enhance the dataset even further. In the subsequent parts, we will cite your specific concerns and provide thorough explanations to address them.
>
> > Make the paper more formal.
> >
>
> Thanks for the suggestion! We have further revised the paper with more formal fonts and descriptions.
>
> > This dataset presents an original contribution to the field.
> >
> 1. **Midjourney is popular and representative.** Midjourney is one of the most popular text-to-image generation agent with more than **16 million** registered users **from all over the world, in diverse age range, with various gender, culture background, ethnicity, etc**. It gets approximately **28.5 million visits** per month, leading to an overwhelming number of **informative and diverse prompts** and **high consistent and high quality images** in our life. Therefore, it is representative and desired to understand and study the generated contents from Midjourney.
> 2. **Easy extension.** Besides the large amount of Midjourney data, our dataset could be easily extend to other models. Inspired by the review feedback, we additionally introduce another **22 text-to-image generative models** to our datasets, such as Dreamlike Photoreal 2.0, DALL·E 2, StableDiffusion-XL, ChilloutMix, etc., which significantly improves the diversity of our JourneyDB, making it a **comprehensive benchmark** for evaluating models’ comprehension of generated images.
> 3. **Large amount of annotations.** Besides scraping the midjourney images, we also provide a **large amount of annotations** with both GPT and human annotators on top of the original prompts. As explored in previous paper, e.g. VideoChat and LLaVA, such annotations generated from GPT also **significantly benefit the instruction tuning**.
> 4. **Generalisation.** We evaluate the models on the **extended test set generated with additional 22 text-to-image generative models**. We find that the  model, **fine-tuned on the Midjourney dataset**, has a **better performance** on this extended test set, which indicates that Midjourney dataset helps models to understand generated images **generally**.
>
> Table A. Image Caption Results on the Additional 22 Text-to-Image Models
>
> | Backbone | OPT | FLAN | Uni-Perceiver v2 | Uni-Perceiver v2 (FT) |
> | --- | --- | --- | --- | --- |
> | METEOR | 9.00 | 9.77 | 5.49 | **9.88** |
> | ROGUE-L | 26.25 | **27.57** | 16.81 | 22.93 |

---

> > ### Author Response · Authors · 2023-08-25
> > **Response to Reviewer J527 (2/2)**
> >
> > > The dataset might contain noise, but we provide clean test set.
> > >
> >
> > Welcome to check the data! We highlight that the **test set**, which includes a lot of manual efforts, is **clean and suitable to be a benchmark**. And thanks to the **detailed and informative prompts**, the **consistency** between the prompt and image is strong. Evaluation on our paper and existing work prove the **effectiveness** of such kind of datasets.
> >
> > 1. **Uncurated examples in supplementary materials.** In the **supplementary material**, we have shown some **uncurated examples**.
> > 2. **Check the data.** Check the data. To facilitate the reviewers further checking the data, we prepare a temporal cache of data on http://101.230.144.196:3337. Welcome to download and check them out. Downloading and using the data means that you agree with the JourneyDB license.
> > 3. **Cleaned test set.** Besides the extensive training set, we have also provided a **clean test set as a evaluation benchmark**. In order to ensure the cleanliness of this benchmark, as outlined in Sec. 3.2, a team of **40 annotators** meticulously assesses the **consistency** between the prompts and the corresponding images. Furthermore, to further ensure the quality of the test set, another group of annotators **scrutinizes the annotations** generated by GPT3.5. Hence, the creation of the benchmark is not solely reliant on an automated generation process; rather, it involves **a significant amount of human effort and involvement**. While the inclusion of GPT3.5 has relieved human beings from the task of content generation, it has not completely replaced human annotators.
> > 4. **Clip Score for Consistency.** We calculate the **Clip Score** for the **JourneyDB**, and **Coco Caption**, a well-known image-text dataset. JourneyDB obtains a Clip Score of **84.67%**, which is **60.36%** greater than that of Coco Caption, indicating that the prompts in our dataset have **high consistency** with the corresponding generated images.
> >
> >     Table A. Comparison between Coco Caption and JourneyDB in terms of text-image consistency (Clip Score) and  information amount (Average Length).
> >
> >     |  | Coco Caption | JourneyDB |
> >     | --- | --- | --- |
> >     | Clip Score | 52.80% | **84.67% (+60.36%)** |
> >     | Average Length | 10.45 | **27.29 (+161.15%)** |
> > 5. **Informative Prompts.** JourneyDB contains **long, informative** prompts, which describe the details of the corresponding image. As shown in the Table A, **JourneyDB** has a **161.15%** longer average length than Coco Caption, indicating existing dataset do not describe image in **such a detailed level**, which makes the dataset helpful in **promoting the ability of visual understanding models**.
> > 6. **Data generated by models are useful.** There are many other datasets that are automatically **constructed with models** such as GPT, which might contain noise but can contribute to **fantastic results**. For example, **VideoChat** [A] creates a dataset by feeding dense captions to **ChatGPT** [B] in temporal order. Similarly, **LLaVA** [C] constructs a instruction-tuning dataset with **GPT4** [D]. **LaCLIP** [E] incorporates text augmentations through text rewriting by **ChatGPT** [B] and **Bard**[F]. And **StableRep** [G] reveals that solely synthetic data generated from text-to-image models can be used to train powerful visual representations. These paper demonstrate that the dataset generated by models, although sounds noisy, can make **great contributions in many tasks.**
> > 7. **Improved Fine-tuning Results.** By **finetuning** models on our training set, the **image-caption** and **prompt-inversion** results are **significantly improved on the testing set**, indicating that the training set is informative and effective to train a model for enhancing visual understanding of the generated images.
> >
> > **Reference**
> >
> > [A] Li, KunChang, et al. "Videochat: Chat-centric video understanding." *arXiv preprint arXiv:2305.06355* (2023).
> >
> > [B] [https://chat.openai.com](https://chat.openai.com/)
> >
> > [C] Liu, Haotian, et al. "Visual instruction tuning." *arXiv preprint arXiv:2304.08485* (2023).
> >
> > [D] https://openai.com/gpt-4
> >
> > [E] Fan, Lijie, et al. "Improving CLIP Training with Language Rewrites." *arXiv preprint arXiv:2305.20088* (2023).
> >
> > [F] https://bard.google.com/
> >
> > [G] Tian, Yonglong, et al. "StableRep: Synthetic Images from Text-to-Image Models Make Strong Visual Representation Learners." *arXiv preprint arXiv:2306.00984* (2023).

---

> ### Author Response · Authors · 2023-08-29
> **Looking forward to your reply!**
>
> We sincerely appreciate your great efforts in reviewing this paper. Your constructive advice and valuable comments really help improve our paper. Considering the approaching deadline, please, let us know if you have follow-up concerns. We sincerely hope you can consider our reply in your assessment, and we can further address unclear explanations and remaining concerns if any.
>
> Once more, we are appreciated for the time and effort you've dedicated to our paper.

---

> > ### Comment · Reviewer_J527 · 2023-08-30
> >
> > Thank you for your comments. I recognize all the points being made above and appreciate the author's effort in a detailed response. However, my concern with the closed-source nature of Midjourney remains. Since the authors are not affiliated with Midjourney (as evidenced by single-blindness - we know the identities of the authors), there is not even a guarantee that during sampling the inherent model behind Midjourney remained the same. Combined with a synthetic caption that is also opaquely generated (e.g. GPT 3.5), these problems fundamentally challenge the basis of a "benchmark" to be published in this venue. I agree that the dataset is of decent quality and some significance and that it may help train better models (also remarking how a lot of the referenced works in this response are also preprints under review), but my score remains the same.

---

> > > ### Author Response · Authors · 2023-08-30
> > >
> > > Thanks reviewer for the comments!  We hope our response can address your concerns.
> > >
> > > > Closed-source nature of Midjourney remains.
> > > >
> > >
> > > Please note that besides the large amount of Midjourney data, we additionally introduce another **22 text-to-image generative models** to our datasets, such as Dreamlike Photoreal 2.0, DALL·E 2, StableDiffusion-XL, ChilloutMix, etc., which significantly improves the diversity of our JourneyDB, making it a comprehensive benchmark for evaluating models’ comprehension of generated images.
> > >
> > > > Combined with a synthetic caption that is also opaquely generated (e.g. GPT 3.5), these problems fundamentally challenge the basis of a "benchmark" to be published in this venue.
> > > >
> > >
> > > As for benchmark purpose, we have provided a clean test set, as outlined in Sec. 3.2, a team of 40 annotators meticulously assesses the consistency between the prompts and the corresponding images. Furthermore, to further ensure the quality of the test set, another group of annotators scrutinizes the annotations generated by GPT3.5.Therefore, the creation of the benchmark **is not solely reliant on an automated generation process**; rather, it involves a significant amount of human effort and involvement. While the inclusion of GPT3.5 has relieved human beings from the task of content generation, it has not completely replaced human annotators.
> > >
> > > **In summary**:
> > > 1. We have addressed the **Closed-source nature of Midjourney** by adding 22 text-to-image generative models as dat source to the current dataset and many of them are **open-source models**.
> > > 2. We want to clarify that the annotations of the benchmark **are not solely reliant on an automated generation process** but were further curated and assessed by human annotators.

---

> > > > ### Comment · Reviewer_J527 · 2023-08-30
> > > >
> > > > Thank you for pointing that out. I've adjusted my score accordingly.

---

> > > > > ### Author Response · Authors · 2023-08-31
> > > > >
> > > > > Thank you for adjusting the review score!! We're pleased that your concerns have been addressed. However, we've noticed that the current rating for our paper is still 'marginally below the acceptance threshold.' If you have further suggestions for enhancing our paper and datasets, please don't hesitate to share. We are committed to working hard to implement your suggestions. Thanks again for your constructive engagement!

---

> > > > > > ### Author Response · Authors · 2023-08-31
> > > > > > **Acknowledgement**
> > > > > >
> > > > > > Dear Reviewer J527,
> > > > > >
> > > > > > We are writing to express our **appreciation** for your valuable time and effort in reviewing our paper. We have **thoroughly revised** our dataset according to your suggestion.
> > > > > >
> > > > > > To summarize, your **main concerns** have been addressed:
> > > > > >  1. **Paper may need better Quality Control:** Human annotators have thoroughly revised the test set, ensuring **our benchmark is reliably clean**. And we have fully demonstrated that our training data, as well as other datasets generated with GPT, is **helpful** despite the potential annotation noise.
> > > > > >  2. **Closed-source nature of Midjourney remains:** We additionally introduce another **22** text-to-image generative models to our datasets, such as Dreamlike Photoreal 2.0, DALL·E 2, StableDiffusion-XL, ChilloutMix, etc., which significantly **improves the diversity** of our JourneyDB, making it a **comprehensive benchmark** for evaluating models’ comprehension of generated images.
> > > > > >
> > > > > > Unfortunately, although your mentioned concerns have been addressed, your rating is still 'marginally below the acceptance threshold', without pointing out further concerns.
> > > > > >
> > > > > > If you still have any outstanding concerns or if there is anything further you would like to discuss regarding our paper, please do not hesitate to **reach out to us by any means**. We would be more than willing to address any comments or queries you may have, **even after the discussion period**.
> > > > > >
> > > > > > Once again, we sincerely appreciate your contribution to the review process. We look forward to any additional feedback you may provide, should you have the opportunity to review our response further.
> > > > > >
> > > > > > Thank you!
> > > > > >
> > > > > > Sincerely, \
> > > > > > JourneyDB Authors

---

### Official Review · Reviewer_Wqw6 · 2023-07-22
**JourneyDB: a 4M dataset of generated images with prompts**

**Rating:** 8
**Confidence:** 4

**Strengths:**

Strengths include
- the largest released dataset for generated image / text
- augmentations of prompts to try and provide more information per image
- evaluation of the datasets using 4 tasks and multiple models

**Additional Feedback:**

Thanks for submitting this work.

**Clarity:**

The paper is reasonably well written.
It could be improved by making some sections shorter and adding more explanation on how the images were initially generated.

**Correctness:**

The dataset construction is described precisely. Statistics and evaluation of the dataset prove its value.

**Documentation:**

No link is provided.

**Ethics:**

No ethical concern.

**Limitations:**

The work could be improved by
- comparing real image evaluation with the generated image ones


**Opportunities For Improvement:**

Limitations include
- The usefulness of the prompt augmentation is not evaluated. It may well be that the raw prompts are better
- the paper claims the understanding of pretraining model of generated images is weak, but the evaluation do not compare results on generated images with results on real images
- the matching of prompt and image hence the quality of the pairs is not computed (could be done with eg clip)

**Relation To Prior Work:**

Text to image models are described. There are many image/text datasets, it may be useful to describe more of them.

**Summary And Contributions:**

This paper introduces
- 4M generated images with prompts scrapped from midjourney discord
- an evaluation of the capacity of models to analyze such generated images: prompt inversion, style retrieval, image captioning, visual question answering
- they use gpt to convert prompts into more specific information (content, style, caption, qa)
- asked 40 annotators to remove unrelated words in the prompts
- they evaluate image captioning by training and testing on their dataset and compare with other models trained on other datasets
- they evaluate the capacity of pretrained CLIP to do style retrieval on their dataset
- they evaluate the capacity of 3 models to do zero shot visual question answering on their dataset

They claim the results obtained on these evaluations show that the understanding of generated images by pretrained models is poor.

---

> ### Author Response · Authors · 2023-08-25
> **Response to Reviewer Wqw6**
>
> #
>
> We sincerely appreciate your positive response to our dataset. Furthermore, we would like to extend our heartfelt gratitude for your invaluable comments. Your concerns hold immense value to us as they are a source of inspiration to continually enhance the dataset. In the subsequent parts, we will cite your specific concerns and provide explanations to address them.
>
> > Evaluation: raw prompts vs. prompt augmentation
> >
>
> We do have an augmentation when constructing the dataset **by removing the inconsistent prompt words and the specific Midjourney flags**. As shown in Table A, we calculate **Clip Score [A]** for these two versions, and notice that the **augmentation of the prompt is slightly better** than that without augmentation.
>
> Table A. Comparison of Clip Score for Prompt w/wo Augmentation
>
> |  | w/o Prompt Augmentation | w/ Prompt Augmentation |
> | --- | --- | --- |
> | Clip Score |  84.60% | **84.67%** |
>
> > Results on real image datasets
> >
>
> We have added a column in the Table2 in the main paper. The performance on real image dataset is significantly better than the performance in our dataset.
>
> > Clip score to compute the pair quality
> >
>
> We calculate the Clip score for the training and testing set. The high clip-score indicates that the prompts in our dataset have high consistency with the corresponding generated images.
>
> > Shorten some sections and explain more on how the images were initially generated.
> >
>
> Thank you for the suggestion! We have removed a redundant image and shorten some sections, and added the related details to the **Sec. 3.1 data collection**.
>
> In the public Discord [B] channel called “Midjourney”, users send the text prompt to the channel, and the Midjourney robot responds with the corresponding generated images. The users choose the one they prefer to upscale, and Midjourney would return the corresponding upscaled images.
>
> > Describe more of the image/text datasets.
> >
>
> Thank you for the suggestion! We have added more introductions to existing image-text datasets in the related works.
>
> **Reference**
>
> [A] https://github.com/jmhessel/clipscore
>
> [B] https://support.discord.com/hc/en-us

---

> > ### Comment · Reviewer_Wqw6 · 2023-08-29
> >
> > Thank you for the detailed answer and the improvements!

---

> > > ### Author Response · Authors · 2023-08-29
> > > **Thanks for your feedback!**
> > >
> > > We are glad to have received your message! Thank you for recognizing our paper. Your advice and comments have truly contributed to the improvement of our work!

---

### Official Review · Reviewer_r9bw · 2023-07-28
**interesting dataset, might need better quality control**

**Rating:** 4
**Confidence:** 4
**Correctness:** mostly correct
**Clarity:** yes.

**Strengths:**

1. the scale of the dataset is impressive.
2. it's interesting to see the creation of such dataset under different styles.


**Additional Feedback:**

1. It unclear to me why the authors choose to use a less common font in Figures. While the visual appearances are nice, the fonts might not be necessarily considered rigorous enough for an academic product.

2. Similarly, Figure 1 is stunning, but it does not seem to belong to a paper.

**Documentation:**

The discussion does not seem sufficient regarding the guideline questions such as "availability and maintenance, and ethical and responsible use". No link is provided, but can be easily tracked through Google search. The paper is submitted in a double-blind format, not sure if the authors know that they can reveal themselves and show the data webpage.

**Ethics:**

Unfortunately, there are concerns that some images might contain offensive content (As the authors admit themselves).

**Limitations:**

The authors admit themselves that "we did not check all images to make sure the content is free of offensive or violent content." Unfortunately, while I appreciate the authors honesty, this issue might be particularly strong, and admitting this issue might not be enough to release the data.

**Opportunities For Improvement:**

1. the quality control of the dataset might need additional work. For example, the answers are generated by GPT at this moment, and some of the questions need further checkings by human.

   - for example, I appreciate the authors' honesty in showing Figure 5, but this will also raise the questions of the dataset. Personally, I feel like ground truth of Q1 and Q4 (and perhaps also Q2 and Q3) are ambiguous. Then, further, it seems Q5 and Q7's ground truth are incorrect. This seems paint a strong issue about the quality of the datasets.


2. while the results shown mostly about zero-shot performances, it's unclear if there are (shortcut) biases in stand-alone models (training/fine-tuning with training set) and then testing with testing set.

**Relation To Prior Work:**

highly relevant: Breaking Common Sense: WHOOPS! A Vision-and-Language Benchmark of Synthetic and Compositional Images

**Summary And Contributions:**

The authors of the paper present a novel large-scale dataset, JourneyDB, created specifically to enhance the visual understanding of generatively created content. This dataset includes 4 million generated images paired with their associated text prompts, providing a diverse spectrum of content and style.

---

> ### Author Response · Authors · 2023-08-25
> **Response to Reviewer r9bw (1/2)**
>
> We extend our gratitude for your valuable comments. Your concerns are greatly appreciated as they inspire us to enhance the dataset even further. In the subsequent parts, we will cite your specific concerns and provide thorough explanations to address them.
>
> > The quality control of the dataset might need additional work.
> >
>
> The data quality is **better** than what was presented in Figure 5. Welcome to check the data! We highlight that the **test set**, which includes a lot of manual efforts, is **clean and suitable to be a benchmark**. And thanks to the **detailed and informative prompts**, the **consistency** between the prompt and image is strong. Evaluation of our paper and existing work prove the **effectiveness** of such kind of dataset.
>
> 1. **Figure 5 results from mistake.** Thanks for pointing out the problem in **Figure 5**. It was a mistake when drawing Figure 5, where the ground truth are all mistakenly set to “C”. The ground-truth label should have been “**B, C, A, B**” for the **bottom image**.
> 2. **Uncurated examples in supplementary materials.** In the **supplementary material**, we have shown some **uncurated examples**.
> 3. **Check the data.** To facilitate the reviewers further checking the data, we prepare a temporal cache of data on [http://101.230.144.196:3337](http://101.230.144.196:3337/). Welcome to download and check them out. Downloading and using the data means that you agree with the [JourneyDB license](https://github.com/JourneyDB/JourneyDB/blob/main/assets/Terms_of_Usage.md).
> 4. **Cleaned test set.** Besides the extensive training set, we have also provided a **clean test set as a evaluation benchmark**. In order to ensure the cleanliness of this benchmark, as outlined in Sec. 3.2, a team of **40 annotators** meticulously assesses the **consistency** between the prompts and the corresponding images. Furthermore, to further ensure the quality of the test set, another group of annotators **scrutinizes the annotations** generated by GPT3.5. Hence, the creation of the benchmark is not solely reliant on an automated generation process; rather, it involves **a significant amount of human effort and involvement**. While the inclusion of GPT3.5 has relieved human beings from the task of content generation, it has not completely replaced human annotators.
> 5. **Clip Score for Consistency.** We calculate the **Clip Score** for the **JourneyDB**, and **Coco Caption**, a well-known image-text dataset. JourneyDB obtains a Clip Score of **84.67%**, which is **60.36%** greater than that of Coco Caption, indicating that the prompts in our dataset have **high consistency** with the corresponding generated images.
>
>     Table A. Comparison between Coco Caption and JourneyDB in terms of text-image consistency (Clip Score) and  information amount (Average Length).
>
>     |  | Coco Caption | JourneyDB |
>     | --- | --- | --- |
>     | Clip Score | 52.80% | **84.67% (+60.36%)** |
>     | Average Length | 10.45 | **27.29 (+161.15%)** |
> 6. **Informative Prompts.** JourneyDB contains **long, informative** prompts, which describe the details of the corresponding image. As shown in the Table A, **JourneyDB** has a **161.15%** longer average length than Coco Caption, indicating existing dataset do not describe image in **such a detailed level**, which makes the dataset helpful in **promoting the ability of visual understanding models**.
> 7. **Data generated by models are useful.** There are many other datasets that are automatically **constructed with models** such as GPT, which might contain noise but can contribute to **fantastic results**. For example, **VideoChat** [A] creates a dataset by feeding dense captions to **ChatGPT** [B] in temporal order. Similarly, **LLaVA** [C] constructs a instruction-tuning dataset with **GPT4** [D]. **LaCLIP** [E] incorporates text augmentations through text rewriting by **ChatGPT** [B] and **Bard**[F]. And **StableRep** [G] reveals that solely synthetic data generated from text-to-image models can be used to train powerful visual representations. These paper demonstrate that the dataset generated by models, although sounds noisy, can make **great contributions in many tasks.**
> 8. **Improved Fine-tuning Results.** By **finetuning** models on our training set, the **image-caption** and **prompt-inversion** results are **significantly improved on the testing set**, indicating that the training set is informative and effective to train a model for enhancing visual understanding of the generated images.
>
> > It's unclear if there are (shortcut) biases in stand-alone models
> >
>
> Yes, we provide results of fine-tuned models.
>
> As introduced in **Sec. 4.1 and 4.2** of the main paper, we **finetuned the Uni-Perceiver v2** on **image caption** and **prompt inversion** tasks on our training set to verify the effectiveness of our data. We have updated the table to highlight the finetuning models.

---

> > ### Author Response · Authors · 2023-08-25
> > **Response to Reviewer r9bw (2/2)**
> >
> > > Violent Contents
> > >
> >
> > We have thoroughly revised the dataset to ensure the dataset free from harmful or violent images.
> >
> > On the one hand, **Midjourney itself has provided a violence filtering** to ensure the presented images and prompts are suitable for the network [H]. On the other hand, we apply a NSFW model [I] on the whole dataset to further analyse the the violence extend of JourneyDB and **manually filter** the images indeed unsuitable for work.
> >
> > Images with the NSFW score [J] higher than 0.8 are likely to be “not-suitable-for-work” images, while the NSFW score smaller than 0.2 indicates the images are highly likely to be “safe-for-work”. JourneyDB gets an average NSFW score of **0.008**, where **99.01%** images get scores **smaller than 0.2 (safe for work)**, and only **0.16%** are greater than **0.8**. We manually go through these images, and **find out and remove around 132 images** indeed unsuitable for work from the dataset.
> >
> > We hope our efforts will contribute to the meticulous cleansing of the dataset, thereby providing a sanitized and non-violent dataset for the community.
> >
> > > Further Discussion on Availability and Maintenance
> > >
> >
> > The dataset is publically available for research purpose only. And authors of this paper will serve as maintainers for this paper. For more details, please refer to the data sheet on **Sec. 4.6 and 4.7** of the **supplementary materials**.
> >
> > > Reveal the Authorship
> > >
> >
> > Thank you for the suggestion! We have reveal the authors in the latest revision.
> >
> > > Make the Paper More Formal
> > >
> >
> > Thanks for the suggestion! We have revised the paper, using more formal fonts and descriptions.
> >
> > **Reference**
> >
> > [A] Li, KunChang, et al. "Videochat: Chat-centric video understanding." *arXiv preprint arXiv:2305.06355* (2023).
> >
> > [B] [https://chat.openai.com](https://chat.openai.com/)
> >
> > [C] Liu, Haotian, et al. "Visual instruction tuning." *arXiv preprint arXiv:2304.08485* (2023).
> >
> > [D] https://openai.com/gpt-4
> >
> > [E] Fan, Lijie, et al. "Improving CLIP Training with Language Rewrites." *arXiv preprint arXiv:2305.20088* (2023).
> >
> > [F] https://bard.google.com/
> >
> > [G] Tian, Yonglong, et al. "StableRep: Synthetic Images from Text-to-Image Models Make Strong Visual Representation Learners." *arXiv preprint arXiv:2306.00984* (2023).
> >
> > [H] https://docs.midjourney.com/docs/community-guidelines
> >
> > [I] https://github.com/LAION-AI/CLIP-based-NSFW-Detector
> >
> > [J] https://github.com/yahoo/open_nsfw

---

> ### Author Response · Authors · 2023-08-29
> **Looking forward to your reply!**
>
> We sincerely appreciate your great efforts in reviewing this paper. Your constructive advice and valuable comments really help improve our paper. Considering the approaching deadline, please, let us know if you have follow-up concerns. We sincerely hope you can consider our reply in your assessment, and we can further address unclear explanations and remaining concerns if any.
>
> Once more, we are appreciated for the time and effort you've dedicated to our paper.

---

> ### Author Response · Authors · 2023-08-31
> **Sincere Request for Further Discussions**
>
> Dear Reviewer,
>
> Thanks again for your great efforts in reviewing this paper! With the discussion period drawing to a close, we expect your feedback and thoughts on our reply. We put a significant effort into our response, with several new adjustments and discussions. We sincerely hope you can consider our reply in your assessment. We look forward to hearing from you, and we can further address unclear explanations and remaining concerns if any.
>
> Best,
>
> Authors

---

> > ### Author Response · Authors · 2023-08-31
> > **Acknowledgment and Understanding**
> >
> > Dear Reviewer r9bw,
> >
> > We are writing to express our **appreciation** for your valuable time and effort in reviewing our paper. We have **thoroughly revised** our dataset according to your suggestion.
> >
> > To summarize, your **main concerns** have been addressed:
> >  1. **Paper may need better Quality Control:** Human annotators have thoroughly revised the test set, ensuring **our benchmark is reliably clean**. And we have fully demonstrated that our training data, as well as other datasets generated with GPT, is **helpful** despite the potential annotation noise.
> >  2. **Some images might contain offensive content:** We thoroughly revised the dataset via **model filtering** and **manual checking** to ensure the dataset does **not contain** such harmful or violent images.
> >
> > Unfortunately, we have **not received** a response from you during this discussion period. We **understand** that circumstances may arise that prevent timely communication, and we **respect** any constraints or commitments you may have encountered. We would acknowledge this situation and express our **regret** that we could not engage in a direct dialogue with you.
> >
> > If you still have any outstanding concerns or if there is anything further you would like to discuss regarding our paper, please do not hesitate to **reach out to us by any means**. We would be more than willing to address any comments or queries you may have, **even after the discussion period**.
> >
> > Once again, we sincerely appreciate your contribution to the review process. We look forward to any additional feedback you may provide, should you have the opportunity to review our response further.
> >
> > Thank you!
> >
> > Sincerely, \
> > JourneyDB Authors

---

### Author Response · Authors · 2023-08-30
**Letter to Area Chair**

Dear Area Chair,

Thank you for your dedicated efforts in organizing the review process for our paper. We sincerely appreciate the time and feedback provided by all reviewers.

**Common concerns raised by the reviewers that have been addressed during the discussion period**:

1. **Objectionable content risk**: We have conducted thorough human and machine-based filtering of the dataset to ensure its integrity and eliminate any objectionable content.
2. **Data source limitation**: We have expanded our dataset by incorporating data generated by **22** text-to-image generative models, enhancing the diversity of our data.
3. **Noisy generated annotation:** Our annotators have thoroughly revised the test set and we have demonstrated that our training data is helpful despite the potential annotation noise.

**Current Reviewing status**:

1. **Wqw6**: Maintains clear acceptance post-discussion.
2. **Q5nG** and **J527**: Raise rating with **positive** feedback.
3. **r9bw** and Ethics Reviewer **em4x**: Share concerns with **Q5nG**, awaiting further feedback.

We value reviewers' feedback and remain open to addressing concerns. Their constructive advice and valuable comments really helped improve our paper. As we approach the deadline, we eagerly anticipate the response from the remaining reviewers. We truly appreciate your support in facilitating productive discussions throughout this process.

Thanks for your time and support.

Sincerely,

JourneyDB Authors

---

### Decision · Program_Chairs · 2023-09-22

**Decision:**

Accept (Poster)

**Comment:**

Key contribution:
This paper introduced a large dataset: 4M generated images with prompts scrapped from midjourney discord
an evaluation of the capacity of models to analyze such generated images: prompt inversion, style retrieval, imagecaptioning, visual question answering.  GPT was used to convert prompts into more specific information (content, style, caption, qa) and humans in the loop were utilized to filter the GPT generated prompts. The dataset was used to evaluate image captioning by model construction and test using their dataset and comparison with models trained on other datasets. The capacity of pretrained CLIP to do style retrieval on their dataset along with the capacity of 3 models to do zero shot visual question answering on their dataset was illustrated.
The central claim is that the results obtained on these evaluations show that the understanding of generated images by pretrainedmodels is poor.

Pros: the largest released dataset for generated image / text, human filtered augmentations of prompts to try and provide more information per image, evaluation of the datasets using 4 tasks and multiple models.

Cons:   A number of concerns were raised by the reviewers including:  the nature of generative AI and the dynamics of generative AI models and what the meaning of this benchmark will be from a scientific perspective. Moreover, the ethical aspects of scraping data from discord were raised as potential issues.

The authors have addressed a lot of the concerns of the reviewers and the ratings were raised.  With the exception of one of the reviewer who gives a high rating (8), two other reviewers have after the dialogue and revision increased their ratings to 5 and 6. A final reviewer who had ethical concerns communicated after the close of the review process that he also increases the rating to a 5 based on the rebuttal and discussions (this unfortunately is not reflected in the rating since he could not update the review in time and private communication was used to ascertain his final opinion).  The overall average rating thus is a 6 (marginally above acceptance threshold).

Significance and Recommendation:   The extent of discussion based on the revision based on the rebuttal has clarified some of the elements raised including ethics, correctness verification of data, etc.  I find that the size of the dataset and the nature of question raised is interesting as generative AI is a rapidly evolving space and it is important to have benchmarks. However, I have reservations on the impact of this dataset and benchmark. As nicely articulated by one of the reviewers the central question we need to ask is what specifically the benchmark’s purpose is from a scientific point of view.  The efforts that went into curation and preparing such a large scale benchmark is to be lauded, but I am afraid that the impact is short-term. Thus, I find that the paper is marginally above acceptance threshold as a Poster.